# Pumilio differentially binds to mRNA 3′ UTR isoforms to regulate localization of synaptic proteins

Dominika Grzejda [1,8,12], Anton Hess[1,2,3,12], Andrew Rezansoff[1], Sakshi Gorey[1,2,3], Judit Carrasco[1,9], Carlos Alfonso-Gonzalez[1,10], Stylianos Tsagkris [1,11], Lena Neuhaus[1,2], Mengjin Shi[1,2], Hasan Can Ozbulut [1,2,3], Friederike-Nora Vögtle [4,5,6], Andreas Vlachos [7] & Valérie Hilgers [1,6✉]

## Abstract

**In neuronal cells, the regulation of RNA is crucial for the spatio-temporal control of gene expression, but how the correct localization, levels, and function of synaptic proteins are achieved is not well understood. In this study, we globally investigate the role of alternative 3′ UTRs in regulating RNA localization in the synaptic regions of the Drosophila brain. We identify direct mRNA targets of the translational repressor Pumilio, finding that mRNAs bound by Pumilio encode proteins enriched in synaptosomes. Pumilio differentially binds to RNA isoforms of the same gene, favoring long, neuronal 3′ UTRs. These longer 3′ UTRs tend to remain in the neuronal soma, whereas shorter UTR isoforms localize to the synapse. In cultured *pumilio* mutant neurons, axon outgrowth defects are accompanied by mRNA isoform mislocalization, and proteins encoded by these Pumilio target mRNAs display excessive abundance at synaptic boutons. Our study identifies an important mechanism for the spatiotemporal regulation of protein function in neurons.**

**Keywords** Pumilio; mRNA Localization; Neuronal 3′ UTR; Synaptosome; Synaptic Proteins
**Subject Categories** Neuroscience; RNA Biology

## Introduction

In neurons, the localization of mRNA molecules within specific subcellular compartments, and their local translation, is essential for the spatial regulation of gene expression, and for a rapid and efficient stimulus response (Sun and Schuman, 2023). Neuron-specific RNA-binding proteins (RBPs) interact with sequence or structure elements, usually located in the mRNA 3′ untranslated

region (3′ UTR), to regulate protein function at multiple levels, including transcript localization, translation, and degradation (Bourke et al, 2023; Mayr, 2017; Mitschka and Mayr, 2022). Most metazoan genes express mRNAs with several 3′ UTR variants, generated by the alternative use of polyadenylation (poly(A)) sites (Gruber and Zavolan, 2019). In animal neurons, the highly conserved RBP Embryonic Lethal Abnormal Vision (ELAV) promotes the synthesis of longer, neuron-specific 3′ UTRs in hundreds of genes (Carrasco et al, 2020; Hilgers et al, 2012; Wei et al, 2020). By providing an additional platform for regulatory factors to bind, these sequences are thought to modify the physiological function of the encoded protein by altering the localization, stability, and plasticity of mRNAs isoforms in neuronal compartments (Tushev et al, 2018) (Bourke et al, 2023). The differential usage of 3′ UTR in neuronal projections vs. cell bodies is wide-spread (Arora et al, 2021; Bauer et al, 2019; Ciolli Mattioli et al, 2019; Goering et al, 2023; Mendonsa et al, 2023; Taliaferro et al, 2016), and functions for distinct 3′ UTR variants have been reported for individual genes. For example, the dendritic targeting of the long 3′ UTR isoform of brain-derived neurotrophic factor (BDNF) is essential for proper dendritic development and long-term potentiation (An et al, 2008), whereas in sympathetic neurons, shortening of the Inositol Monophosphatase 1 (IMPA1) 3′ UTR by cleavage is required for maintaining axon integrity (Andreassi et al, 2021).

Binding motifs for the RBP Pumilio (Pum) were found enriched in the neuron-specific 3′ UTR sequences of Drosophila mRNAs (Hilgers et al, 2011; Sanfilippo et al, 2017), raising the possibility that specific RBPs are involved in the global, isoform-dependent modulation of mRNA behavior in neurons. Pumilio proteins are highly conserved members of the PUF (pumilio and fem-3 mRNA-binding factor) family (Goldstrohm et al, 2018). The gene was discovered in Drosophila melanogaster, where, in addition to roles in embryonic patterning (Lehmann and Nüsslein-Volhard, 1987), Pum regulates stem cell self-renewal (Gilboa and Lehmann, 2004; Wang and Lin, 2004), synaptogenesis, neuronal excitability, synaptic excitability and plasticity, and circadian rhythms (Menon

[1]Max-Planck-Institute of Immunobiology and Epigenetics, 79108 Freiburg, Germany. [2]Faculty of Biology, Albert-Ludwigs-University, 79104 Freiburg, Germany. [3]International Max Planck Research School for Immunobiology, Epigenetics and Metabolism (IMPRS-IEM), Freiburg, Germany. [4]Center for Molecular Biology of Heidelberg University (ZMBH), Im Neuenheimer Feld 282, 69120 Heidelberg, Germany. [5]Aging Research, Heidelberg University, 69120 Heidelberg, Germany. [6]Signalling Research Centre CIBSS, University of Freiburg, Schänzlestraße 18, 79104 Freiburg, Germany. [7]Department of Neuroanatomy, Institute of Anatomy and Cell Biology, Faculty of Medicine, University of Freiburg, Freiburg, Germany. [8]Present address: Neuroscience and Rare Diseases, Roche Pharma Research and Early Development (pRED), Roche Innovation Center Basel, Basel, Switzerland. [9]Present address: Discovery Sciences, BioPharmaceuticals R&D, AstraZeneca, CB2 0AA Cambridge, UK. [10]Present address: Genome Biology Unit, European Molecular Biology Laboratory (EMBL), Heidelberg, Germany. [11]Present address: Epigenetics & Neurobiology Unit, European Molecular Biology Laboratory (EMBL), Rome, Italy. [12]These authors contributed equally: Dominika Grzejda, Anton Hess. ✉E-mail: hilgers@ie-freiburg.mpg.de

et al, 2009; Muraro et al, 2008; Schweers et al, 2002). In mammals, Pum 1 and 2 are involved in a variety of important nervous-system-related processes such as neurogenesis, axon guidance, membrane excitability and synaptic plasticity (Dong et al, 2018; Driscoll et al, 2013; Vessey et al, 2010). Molecularly, Pumilio proteins regulate gene expression at the post-transcriptional level by binding to specific sequences in the 3′ UTR of target mRNAs (Jarmoskaite et al, 2019; Zhang et al, 2017). One well-described consequence of Pum binding is the translational repression of target mRNAs, often in conjunction with transport and localized protein synthesis (Goldstrohm et al, 2018; Nishanth and Simon, 2020).

Here, we show that 3′ UTR diversity regulates the expression of protein-coding genes in synaptic regions of the Drosophila brain. We biochemically isolate synaptosomes—structures that contain presynaptic terminals, synaptic vesicles, and often a portion of the postsynaptic membrane, and find that hundreds of mRNAs express 3′ UTR variants that are differentially localized in neuronal subcompartments. We identify direct mRNA targets of Pum in head tissue and find that the role of Pum family proteins in binding synaptic mRNAs is conserved, with many common target genes in flies and mammals. Pum binds to long, neuron-specific 3′ UTRs to promote mRNA localization and regulate the expression of the encoded protein in synaptic compartments. Our results reveal a mechanism of 3′ UTR-dependent regulation, demonstrating how neurons can achieve high spatial protein complexity with a restricted set of genes.

# Results

## Drosophila Pumilio directly binds mRNAs encoding synaptic proteins

We undertook a systematic characterization of mRNAs regulated by Pum binding in the adult Drosophila brain. We C-terminally tagged the endogenous *pumilio* locus with Flag-HA (Fig. EV1A,B). Head tissue of the thus-generated *pum*$^{Flag}$ flies was used to perform RNA immunopurification with UV-cross-linking (xRIP) and recover RNAs directly bound by Pum in vivo (Fig. 1A). We also performed Flag IP on flies that do not express an endogenous tag, in order to control for unspecific binding. We eluted the Pum-bound RNAs and sequenced them by 3′-end sequencing (3′-seq), a method that allows for quantitative measurements of mRNAs with distinct 3′-end isoforms. 3′-seq reads were mapped, filtered for aberrant 3′-ends, clustered and quantified at the 3′-end cluster level (Fig. 1A).

We identified 460 genes whose mRNAs were highly and specifically enriched in the Pum IP compared to the input sample and to the control IP (Figs. 1B,C and EV1C, Dataset EV1); we hereafter refer to these genes and corresponding transcripts as "Pum targets" and "Pum target mRNAs", respectively. Consistent with a highly efficient and specific Pum xRIP, the reported RNA-binding motif for Drosophila Pum (Pumilio Response Element (PRE) (Ray et al, 2013)) constituted the top-enriched motif in 3′ UTR sequences of Pum-bound mRNAs (Fig. EV1D, Dataset EV1), and we recovered the mRNA encoding the sodium channel paralytic (para), a validated functional and physical interactor of Pum in the nervous system (Muraro et al, 2008). Pum targets include numerous genes with a reported role in synaptic transmission (e.g., *Dopamine 1-like receptor 1* (*Dop1R1*),

*Synaptotagmin 1* (*Syt1*), *Vesicular acetylcholine transporter* (*VAChT*), nicotinic acetylcholine receptor subunits (*nAChRα1*, *nAChRβ1,* and *nAChRα6*), memory (e.g., *klingon* (*klg*), *orb2*, *dunce* (*dnc*), *rutabaga* (*rut*)), and genes involved in phenotypes of neuronal excitability (e.g., *shaker cognate w* (*Shaw*), *para*); gene ontology terms for Pum target mRNAs were highly enriched in synaptic processes—most prominently, chemical synaptic transmission (Fig. 1D, Dataset EV1). Interestingly, a substantial fraction of Pum targets represent homologs of genes bound by mouse homologs Pum 1 and/or Pum 2 in neonatal mouse brains (Zhang et al, 2017), and many of those mRNAs were bound by both mouse Pumilios (Figs. 1E and EV1E). This group, which also included Pum itself, contained genes involved in the positive regulation of transcription (Fig. 1F, Dataset EV2). This is consistent with Pum's described role in modulating nuclear effectors including transcription factors (Bohn et al, 2018; Elguindy and Mendell, 2021; Goldstrohm et al, 2018; Wreden et al, 1997). In addition, Gene Ontology (GO) terms were enriched for specialized synaptic functions such as learning and memory, and chemical synaptic transmission (Fig. 1F, Dataset EV2). In conclusion, our data reveal the molecular target mRNAs of Pum in the Drosophila brain, and strongly suggest a conserved role for Pum in the translational regulation and/or localization of mRNAs important for synaptic function.

## Proteins of the Drosophila synaptosome are encoded by Pumilio target mRNAs

Next, we aimed to assess whether Pum was involved in the synaptic localization of its target mRNAs and/or the proteins they encode. Several studies have separated neuronal compartments and assessed differential localization of mRNAs in neurites or synaptic regions of mammalian brains (Cajigas et al, 2012; Tushev et al, 2018) or cultured neurons (Gumy et al, 2011; Mendonsa et al, 2023; Taliaferro et al, 2016; Zappulo et al, 2017); however, such experiments have not been performed in Drosophila. To determine the subcellular distribution of neuronal mRNAs and proteins, we prepared synaptosomes from wild-type adult fly heads. We modified the protocol for biochemical fractionation of neuronal cell compartments from frozen fly head tissue (Depner et al, 2014). Crucially, to preserve RNA/protein complexes, and to exclude content from nuclei—which, in our hands, burst upon homogenization of thawed material—, we employed freshly collected, never-frozen, fly heads. We used homogenization and differential centrifugation with the aim of isolating a fraction enriched in presynaptic and postsynaptic components (Fig. 2A). By transmission electron microscopy, we verified the isolation of intact synaptosomes composed of presynaptic membranes enclosing synaptic vesicles and mitochondria, and adjacent to postsynaptic densities (Fig. 2B). Western Blot analysis confirmed the enrichment of synaptic markers and the depletion of cytoplasmic and nuclear markers (Fig. 2C). Moreover, intronic regions of mRNAs, hallmarks of nuclear pre-mRNAs, were severely depleted in synaptosome fractions (Fig. EV2A). We performed a proteomics analysis of Drosophila synaptosomes as well as crude head homogenate (input) using shotgun mass spectrometry (Dataset EV3). Gene Ontology analysis of 989 proteins enriched in synaptosomes showed that our fraction contained a substantial number of synaptic cellular components, with a strong enrichment of highly specific biological processes related to synaptic function, including

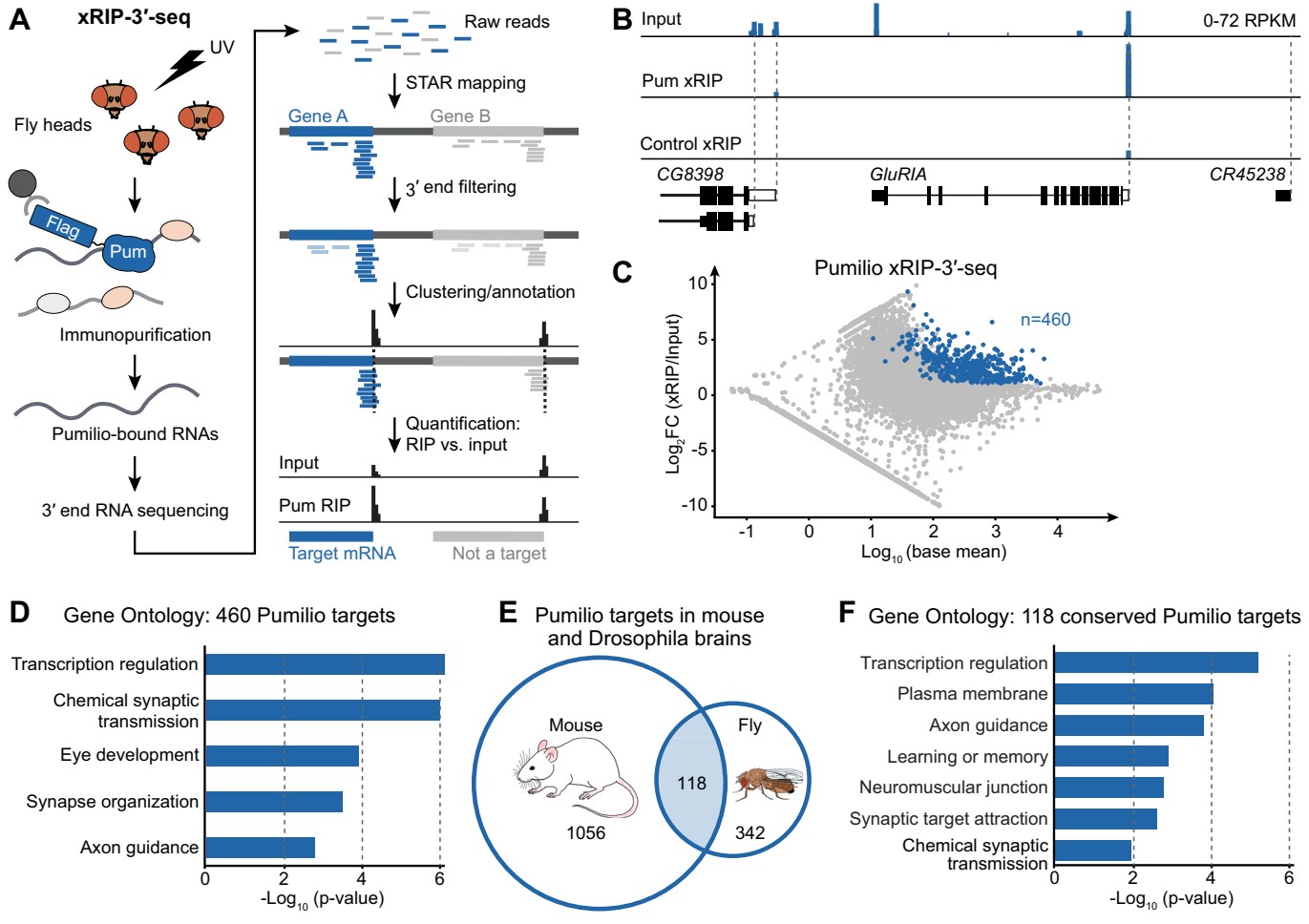

**Figure 1. Pumilio directly binds mRNAs encoding synaptic proteins.**

(A) xRIP-3′-seq experimental and data analysis workflow. Flag-HA-tagged Pum protein was immuno-purified together with UV-cross-linked target mRNAs from adult fly heads, using beads conjugated with an anti-Flag antibody. RNA was subjected to 3′-end sequencing. To determine the exact 3′-end position, sequencing reads were mapped and unreliable clusters were filtered out (see methods). Pum target mRNAs were identified by enrichment of 3′-seq signal in the xRIP sample over input. (B) Visualization of 3′-seq signal for an exemplary Pum target mRNA (*GluRIA*) and the neighboring two non-target genes. Control xRIP represents samples from flies not expressing Flag-tagged Pum ($w^{1118}$). (C) Differential RNA expression in the Pum xRIP sample compared to input (head homogenate), represented as a function of mean expression levels. Dark blue represents: |log$_2$ fold change (Pum xRIP/input)| >1 with *p*-value < 0.05 and base mean >10, and |log$_2$ fold change (Pum xRIP/control xRIP)| >1. (D) Gene ontology analysis of 460 Pum target genes. The top five terms are shown. See Dataset EV1 for all significant terms ($p < 0.05$; one-sided EASE score adjusted using the Benjamini-Hochberg method). (E) Overlap between mouse (Zhang et al, 2017) (Zhang et al, 2017) and Drosophila (this study) Pum target genes. 118 genes represent Drosophila homologs of mouse Pum targets that also figure among the 460 Drosophila Pum targets. (F) Gene ontology analysis of 118 Pum target genes shared between fly and mouse. Significant terms related to nervous system function are shown. See Dataset EV2 for all significant terms ($p < 0.05$; one-sided EASE score adjusted using the Benjamini-Hochberg method).

chemical synaptic transmission and neurotransmitter secretion (Fig. 2D, Dataset EV3). Together, these results show that our neuronal fractionation protocol yields synaptosomes highly enriched in soluble and membrane-bound synaptic proteins.

Strikingly, almost half of all Pum target mRNAs whose encoded protein was detected by mass spectrometry, encode a synaptosome-enriched protein (Figs. 2E and EV2B). Unlike the bulk of Pum target mRNAs, many of which encode proteins involved in transcriptional regulation, this "synaptic" subset was exclusively enriched in functions related to synaptic transmission, neurite guidance and complex behaviors (Fig. 2F, compared to Fig. 1D, Dataset EV3). Those mRNAs stood out by their particularly long 5′ UTRs and 3′ UTRs (over twice as long as non-Pum-bound mRNAs

encoding synaptosome proteins, Fig. EV2C). Longer 3′ UTRs confer added potential for post-transcriptional regulation; for example, Pum target mRNAs encoding synaptic proteins displayed an accumulation of predicted microRNA binding sites (Dataset EV3). Our results may suggest that Pum plays a global and wide-reaching role in the synaptic localization and/or local translation of mRNAs encoding synaptic proteins.

## Synaptically localized mRNAs are not bound by Pum

By 3′-seq, we identified mRNAs of 211 and 1195 genes significantly enriched and depleted in synaptosome fractions, respectively, compared to head homogenate (padj<0.05; Dataset EV3). The

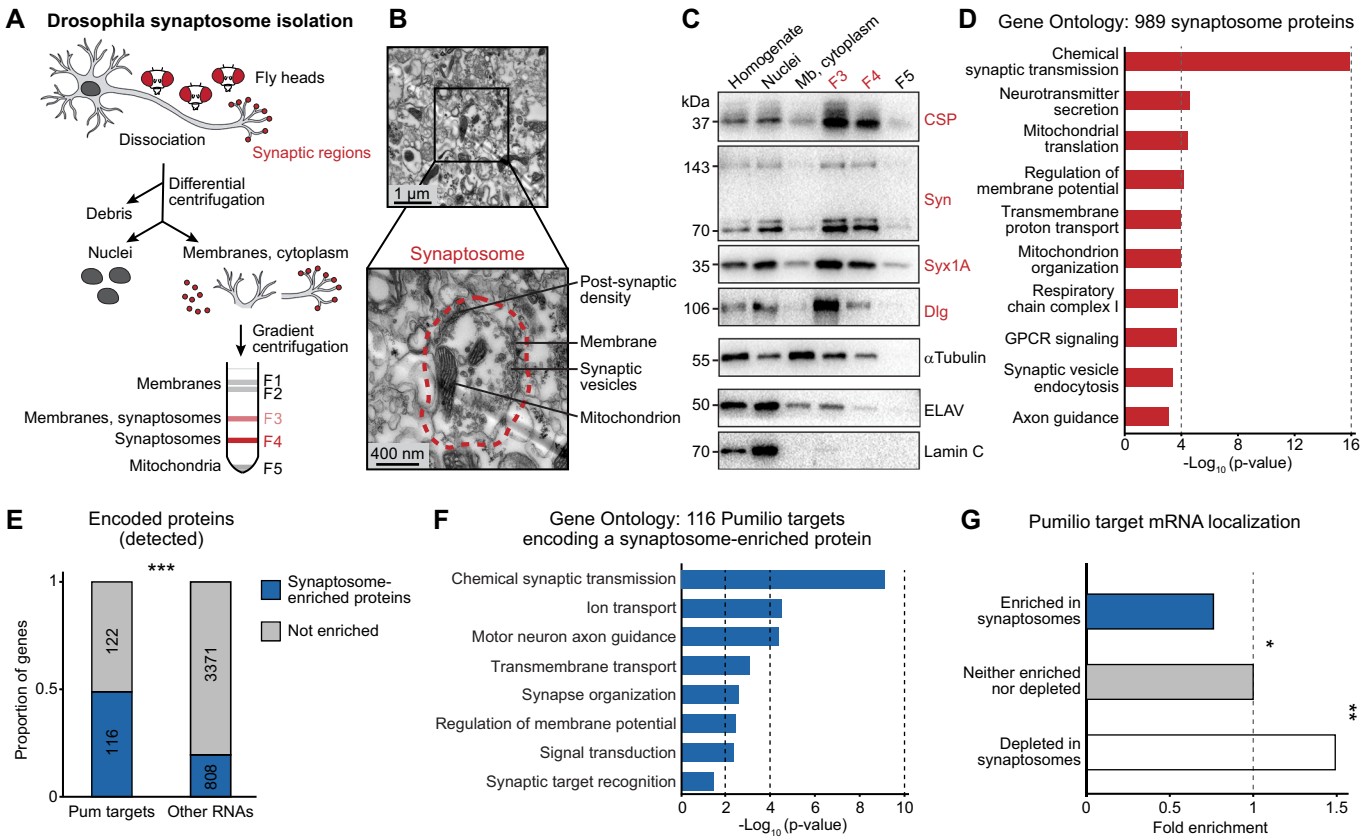

**Figure 2. Pumilio target mRNAs encode proteins enriched in synaptosome fractions.**

(A) Workflow of Drosophila synaptosome isolation. Fresh head tissue from adult Drosophila was dissociated and shear force was used to separate pre- and postsynaptic membranes from nuclei, cytoplasm, and cellular debris. Synaptosome fractions (F3 and F4, red) were enriched by centrifugation on a percoll gradient. (B) Electron micrograph of the synaptosome fraction. The magnification shows an intact synaptosome (indicated by the red border) with its typical components. (C) Western blot showing the expression of the indicated protein markers in the individual fractions. Mb: membranes. Cysteine string protein (CSP), Synapsin (Syn), Syntaxin 1A (Syx1A), and Discs Large (Dlg) represent synaptic markers (in red); ELAV and Lamin C are markers of the nucleosol and nuclear lamina, respectively. (D) Gene ontology analysis of 989 proteins enriched in the synaptosome fraction. The top ten terms are shown. See Dataset EV3 for all significant terms (p < 0.05; one-sided EASE score adjusted using the Benjamini-Hochberg method). (E) Proportion of mRNAs that encode synaptic proteins in each category; "other RNAs" refers to mRNAs that are not bound by Pum and that encode proteins detected by proteomics. ***p < 0.001 (p = 4e−23, two-tailed Fisher's exact test). Only RNAs that encode proteins detected in the synaptosome isolation experiment were considered (see Fig. EV2B). (F) Gene ontology analysis of 116 Pum target mRNAs that encode a protein found enriched in the synaptosome fraction. The top eight terms are shown. See Dataset EV3 for all significant terms (p < 0.05; one-sided EASE score adjusted using the Benjamini-Hochberg method). (G) Proportion of Pum target mRNAs in each category of mRNA subcellular localization, represented as enrichment over non-localized genes. *p < 0.1, **p < 0.05 (two-tailed Fisher's exact test). Source data are available online for this figure.

subset of synaptosome-enriched transcripts, although not yielding any significant GO terms (likely due to low detection), contains multiple mRNAs encoding proteins well-known for specialized neuronal and/or synaptic functions, such as Ankyrin 2 (Ank2), nAChRβ3, Longitudinals lacking (Lola) and Neurotrophin 1 (NT1) (Fig. EV2D). On the other hand, consistent with mRNAs localizing in the subcellular region in which they are translated, synapse-depleted mRNAs encoded proteins involved in broad cellular functions in cell somata and nuclei, predominantly transcription and chromatin organization (Fig. EV2E).

Surprisingly, only seven Pum target mRNAs figured among synaptosome-enriched transcripts, constituting an actual depletion at the synapse; in contrast, Pum target transcripts were enriched among synaptosome-depleted transcripts (Figs. 2G and EV2F,G, Dataset EV3). Hence, mRNAs bound by Pum tend to localize in the somatic compartment of neurons, indicating that Pum binding may

cause mRNAs to be retained in the soma rather than being transported to the synapse. Although unexpected considering the enrichment of genes encoding synaptic proteins among Pum target genes, these results are consistent with findings from cultured mammalian neurons: PREs are overrepresented in the transcriptome of cell bodies compared to axons in mouse, and Pum2 knockdown in rat neurons increased axonal localization of the Pum2 target L1cam and PRE-containing mRNAs (Martínez et al, 2019; Minis et al, 2014). Further considering our finding that among genes encoding synaptosome proteins, Pum targets exhibit a much longer 3′ UTR, we hypothesize a conserved role for Pum proteins in the isoform-specific regulation of protein expression: Pum binding to the longer mRNA isoform of synaptic genes may inhibit their translation and retain them in the soma, whereas the short 3′ UTR isoform may be selectively transported to the synaptic compartment and be expressed locally (Fig. 6).

## Pum binds to specific 3′ UTR isoforms of localized mRNAs

To test this hypothesis, we analyzed the expression of 3′-end variants of the same gene and compared their respective 3′ UTR lengths in synaptosomes and input (Fig. 3A). For each gene with at least two expressed 3′-end isoforms (alternatively polyadenylated: APA genes), we assessed the expression of each isoform and calculated an "average 3′ UTR length", which we represent as a percentage of the length of the longest 3′ UTR. We interpret differences in these expression-weighted 3′ UTR lengths between synaptosome and input as differential localization of mRNA isoforms of a given gene (Fig. 3B). We found 542 genes with significantly localized 3′ UTR isoforms (Fig. 3B,C, Dataset EV3). Notably, Pum target genes were strongly enriched among the 315 genes with shorter 3′ UTRs at the synapse (Figs. 3B–D and EV3A): 43 genes displayed a significantly lower 3′ UTR length-percentage in the synaptosome fraction, with only 10 showing the opposite trend (Fig. 3E, Dataset EV4). Using Pum xRIP-3′-seq data, we assessed binding of Pum to each differentially localized 3′ UTR isoform, and found that Pum generally binds more to the long 3′ UTR isoform than to the short (Figs. 3F and EV3B). Taken together, the integration of Pum xRIP-3′-seq and synaptosome RNA and protein analysis shows that Pum globally binds to long 3′ UTR isoforms of synaptic genes; these long mRNAs are specifically de-localized from synaptic regions. Our results provide further evidence for the model that Pum inhibits long 3′ UTR isoforms of synaptic genes in the soma, thereby promoting the expression of their short counterpart at the synapse.

## Pum-mediated regulation of neuron-specific 3′ UTRs

mRNA isoforms with longer 3′ UTRs, through their additional propensity for binding to effectors—such as Pum—may confer a context- or tissue-specific function to the encoded protein (Mitschka and Mayr, 2022). In light of our findings, we wondered whether Pum may confer additional functions to proteins in a neuron-specific manner. In animals from flies to humans, the inhibition of proximal poly(A) sites specifically in neurons, in favor of more distal ones, generates extended, neuronal 3′ UTRs (nUTRs) in hundreds of genes, (Hilgers et al, 2011; Miura et al, 2013; Smibert et al, 2012; Ulitsky et al, 2012). In Drosophila, nUTRs are mediated by the pan-neuronal protein ELAV (Hilgers et al, 2012). nUTR-containing genes tend to perform specialized neuronal functions, most notably synaptic transmission and complex behaviors (Fig. EV4A, Dataset EV4); interestingly, nUTRs are enriched in PREs (Hilgers et al, 2011; Smibert et al, 2012) compared to their short UTR (sUTR) counterpart (Fig. 4A).

Comparing our xRIP-3′-seq data from fly brains, we found that Pum extensively and specifically binds to this particular set of mRNAs: 71 out of 271 (26%) validated nUTR-containing genes (Carrasco et al, 2020) figure among direct mRNA targets of Pum (Figs. 4B and EV4B, Dataset EV4). Gene ontology analysis on these 71 genes compared to other neuron-expressed genes revealed chemical synaptic transmission as the only significant term (Fig. 4C, Dataset EV4), indicating that this subset of Pum target genes functions in neuron-specific pathways, including synaptic signaling (compare Fig. 1D and Fig. 4C). Consistent with this idea, nUTR-containing genes were highly enriched among genes with localized

3′ UTR isoforms (48 out of 542 mRNAs with differential 3′ UTR length in synaptosomes, Fig. EV4C), and Pum bound more to the long (nUTR-containing) isoform; specifically, they constituted a substantial subset (33%) of Pum target genes for which the short 3′ UTR isoform is synaptically localized. Similarly to other localized 3′ UTR isoforms (Fig. 3), and to an even greater extent, nUTR (long) isoforms were depleted from synaptosomes; we found Pum specifically binding the somatically-localized isoforms of this subset of genes (Figs. 4D and EV4C,D). We propose that by providing a binding platform for Pum, nUTRs, in the neuronal soma, restrict translation of mRNAs encoding synaptic proteins. Since the short isoform of the same gene can localize to the synapse, nUTRs indirectly promote a more localized expression of synaptic proteins.

## Impaired neurite outgrowth, mRNA delocalization, and synaptic protein overexpression in Δpum neurons

To functionally test the involvement of Pum in the localized expression of neuronal mRNAs in vivo, we measured molecular and physiological outcomes of mutating pum in flies. We used a combination of two lack-of-function mutations that each abolish Pum RNA-binding activity, $pum^{ET7}$ and $pum^{ET9}$ (Forbes and Lehmann, 1998). The resulting Δpum animals are subvital due to severe developmental defects and typically die before eclosion (Forbes and Lehmann, 1998; Menon et al, 2004); therefore, we were unable to perform synaptosome purifications on these mutants. As an alternative, we prepared primary cultures of developing neurons from dissected larval brains. On a coated substrate, the cells elaborate complex neurite arbors over the course of several days in vitro. We grew the cells on coverslips for analysis of neuron morphology and expression of mRNA isoforms per gene; we also employed coated microporous membrane inserts for neurites to extend towards the bottom side of the membrane, which allows for their physical separation from cell bodies (Figs. 5A and EV5A,B).

By five days in vitro, differences between wild-type and Δpum neurons were evident by fluorescent microscopy (Fig. 5B). By day 7, the proportion of ELAV-expressing cells was decreased, indicating that loss of Pum specifically affects neuron survival (Fig. EV5C). Moreover, neurite outgrowth was severely reduced, both in terms of neurite length and branching complexity (Figs. 5B,C and EV5D, E). We confirmed that Pum protein is expressed in somata in this in vitro neuronal system, and that for most genes investigated, long 3′ UTR isoforms represent a substantial fraction of total mRNA levels (>50% on average, Fig. EV5F,G). We investigated the subcellular localization of neuronal Pum target mRNAs using qPCR on somatic and neuritic fractions. In Δpum neurons, all RNAs we measured were shifted towards cell bodies, including two Pum-independent, localized RNAs, Arc1 and mimi (Fig. EV5H), which is a likely consequence of the observed loss of neurite structures—and their content—upon Pum loss. In addition, the scarcity of material collectible from Δpum neurites rendered the detection of long 3′ UTR isoforms unreliable; these circumstances precluded a direct comparison of 3′ UTR isoform localization in neurites vs. soma. Notably, we did find, for most genes with the largest 3′ UTR difference between soma and synaptosome (Fig. 3E), that the relative abundance of long 3′ UTR isoforms was disrupted in separated cell bodies of Δpum mutant neurons (Fig. 5D). Finally, we assessed whether the localized, nUTR-dependent Pum binding to target mRNAs affected local expression of the encoded proteins.

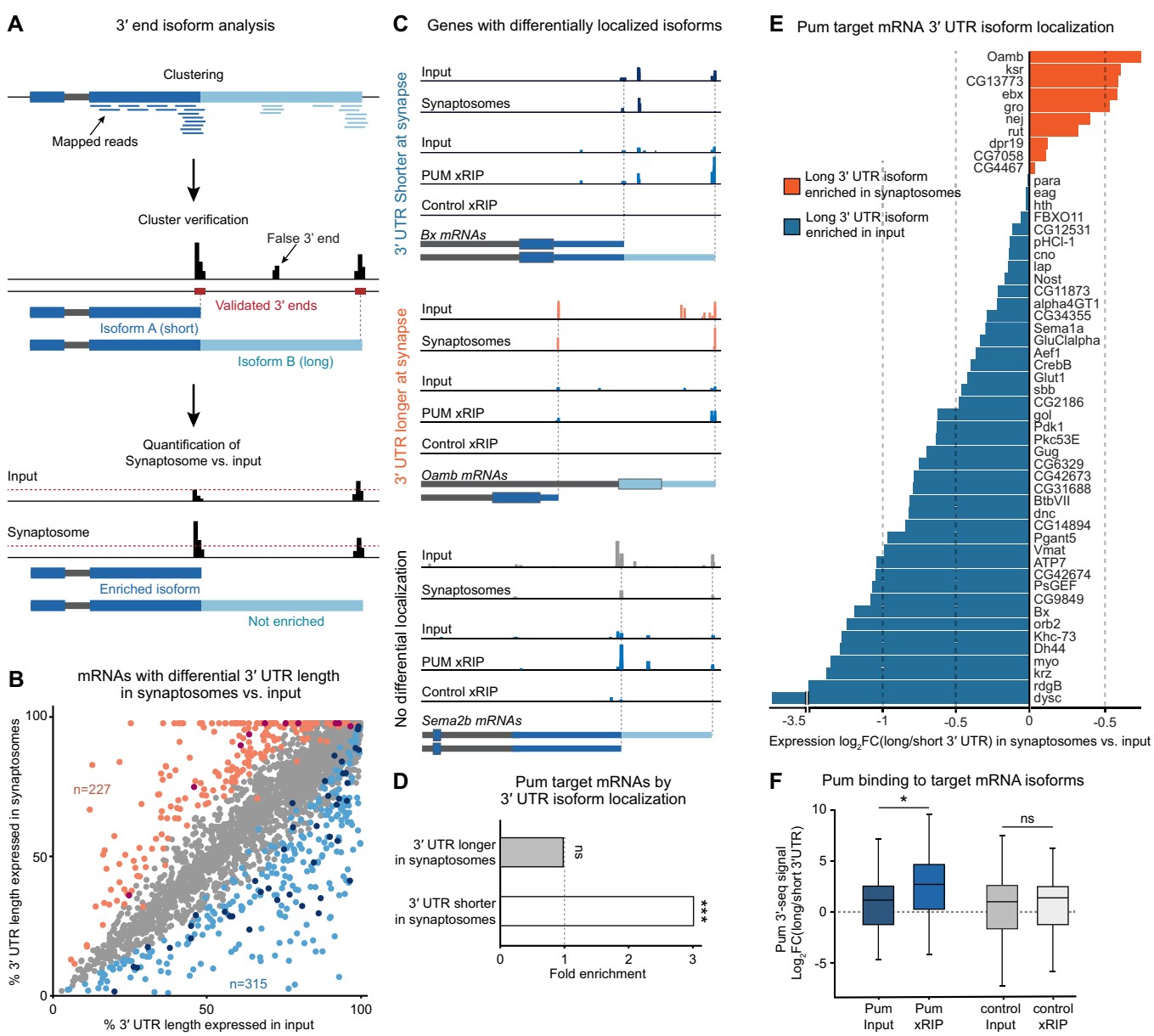

**Figure 3. Synaptic localization of short 3′ UTR isoforms of Pumilio target mRNAs.**

(A) Workflow of differential 3′-end isoform analysis. After mapping and clustering, 3′-seq read clusters were filtered against 3′-ends validated by long-read direct-RNA sequencing (see methods). For each gene, 3′-seq signal was scored for individual clusters in synaptosome fractions and compared to input to identify synaptosome-enriched 3′ UTR isoforms of each gene. (B) Differential 3′ UTR lengths in synaptosomes compared to input. For each gene, the length of each expressed 3′ UTR isoform was represented as a % of the length of the longest 3′ UTR, weighed by expression of each isoform, and combined into one % 3′ UTR length value for each gene. Genes with significant 3′ UTR length differences between synaptosome and input fractions are depicted in orange (longer 3′ UTR in synaptosomes) and blue (shorter 3′ UTR in synaptosomes) (p < 0.05, 1-tailed Z-score). Pum target genes are marked in the darker shade of color. (C) Representative examples (3′-seq tracks and 3′-end isoform models) of genes with differentially localized 3′ UTR isoforms. The upper two traces in each panel show enrichment of the shorter *Beadex* (*Bx*) 3′ UTR isoform (blue), of the longer *Octopamine receptor in mushroom bodies* (*Oamb*) 3′ UTR isoform (orange), or no change for *Semaphorin 2b* (*Sema2b*) in synaptosomes compared to input. The lower three traces show Pum preferential binding to the long (*Bx*, *OamB*) or short (*Sema2b*) 3′ UTR isoforms, respectively. (D) Proportion of Pum target genes in each category of 3′ UTR isoform subcellular localization, represented as enrichment over all expressed genes. ns, non-significant, ***p < 0.001 (two-tailed Fisher's exact test). (E) Enrichment of long vs. short 3′ UTR isoform in synaptosomes compared to input. Shown are Pum target mRNAs from Fig. 3B that display differentially localized 3′ UTR isoforms. 10 and 43 genes display longer 3′ UTR isoforms in synaptosomes (orange) and input (blue), respectively. (F) Pum xRIP-3′-seq signal in long vs. short 3′ UTR isoforms, for the 43 genes in Fig. 3E that display longer 3′ UTR isoforms in the input sample compared to synaptosome sample. ns, non-significant, *p < 0.05; p(Pum Input vs. Pum xRIP) = 0.042; p(control Input vs. control xRIP) = 0.632 (two-tailed Student's t-test). Boxes indicate range between minimum and maximum, the central line depicts the median, lower and upper bounds represent the first and third quartiles, respectively. Control xRIP represents samples from flies not expressing Flag-tagged Pum (*w^1118*). Biological replicates n = 3.

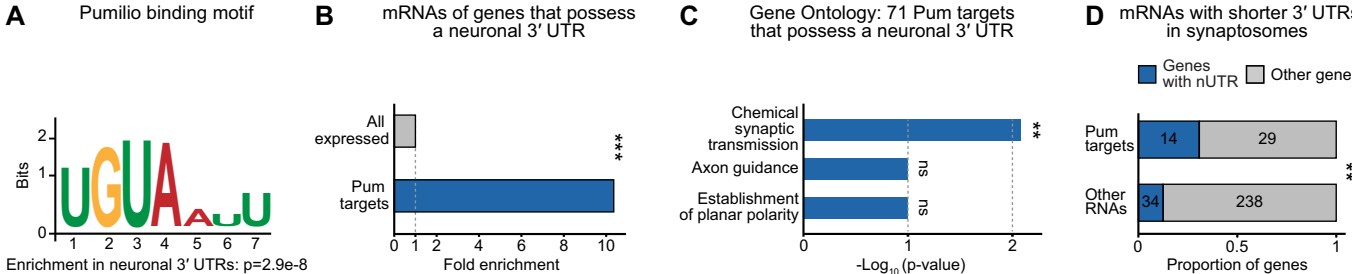

**Figure 4. Pumilio binds to soma-localized neuronal 3′ UTRs of synaptic genes.**

(A) Pum binding motif found enriched in neuronal 3′ UTRs. (B) Proportion of nUTR-containing genes in the indicated gene categories, represented as enrichment over all expressed genes. ***$p < 0.001$ (two-tailed Fisher's exact test). (C) Gene ontology analysis of 71 Pum target genes that possess a neuronal 3′ UTR. The top three terms are shown but only one is significant. **$p < 0.01$ (one-sided EASE score adjusted using the Benjamini-Hochberg method). ns, non-significant. See Dataset EV4 for all terms. (D) Number and proportion of nUTR-containing genes among mRNAs with a shorter 3′ UTR in synaptosomes, in each gene category. *$p < 0.01$ ($p = 0.0012$, two-tailed Fisher's exact test). Only mRNAs expressed in both the synaptosome and input samples were considered.

Antibodies were only available for a subset of them, most of which did not produce reliable localized signal in synaptic boutons of larval neuromuscular junction (NMJs). We did successfully visualize two synaptic proteins, Bruchpilot (Brp) and Syt1 (Synaptotagmin 1). *Syt1* is a conserved, nUTR-regulated Pum target mRNA localized to cell bodies, with the encoded protein enriched in synaptosomes, while *brp*, a Pum target mRNA encoding a synaptic marker, is present but not enriched in synaptosomes (Datasets EV1–EV4). At NMJs of *Δpum* larvae, Brp and Syt1 proteins were significantly overexpressed, with Syt1 exhibiting a particularly drastic increase in abundance (Fig. 5E). This could indicate that localization and translation of Pum target mRNAs are disrupted in the absence of Pum. Although these observations, stemming from developing and in vitro cultured neurons, may not fully reflect RNA processes that occur in the adult brain, they are consistent with our mechanistic model. Overall, our results provide evidence for a role for Pumilio in the localized expression of synaptic proteins in an isoform-specific, nUTR-dependent manner (Fig. 6).

## Discussion

A more complete understanding of neuron-specific RNA regulation is necessary to fully comprehend the complexity of synaptic function. In our study, we reveal one strategy that neurons use to target synaptic protein function to specific cellular compartments: the 3′ UTR-dependent localization of distinct mRNA isoforms of multi-UTR genes, by the RBP Pumilio. We show that in Drosophila brains, the translational repressor binds to distal, often neuron-specific, 3′ UTR regions of mRNAs encoding synaptic proteins, and that these long mRNA isoforms tend to localize to cell bodies. The interaction with Pum likely leads to the retention and translational repression of nUTR isoforms of synaptic genes in somata, while short-UTR variants tend to localize to distal neuronal compartments. Although we did not directly demonstrate local translation of short isoforms at the synapse, the scenario is highly consistent with the observed enrichment of Pum target genes among synaptosome-enriched proteins.

The specific localization of mRNAs to distal neuronal compartments is mediated through 3′ UTR sequence elements that are

recognized and bound by neuronal RBPs including ZBP1 (Ross et al, 1997), FMRP (Goering et al, 2020), Staufen 2 (Bauer et al, 2019), and HBS1L (Mendonsa et al, 2023). Since longer 3′ UTRs contain more RBP binding sites and may allow increased interactions with such regulatory factors, this has led to the general view that the additional UTR sequences mediate specialized mRNA transport to neurites and synaptic regions (Mitschka and Mayr, 2022). Individual long 3′ UTR isoforms have indeed been reported to be targeted towards neuronal projections (An et al, 2008; Andreassi et al, 2010; Perry et al, 2012). While we find that some Pum target genes display a similar regulation (Fig. 3), i.e., localization of the nUTR isoform to synaptosomes, our study suggests that the converse mechanism is more widely used: a specific repression of long isoforms to favor synaptic localization and functionality of their short counterpart. In rat neurons, Pum2 retains specific mRNAs in cell somata, a process important for axonogenesis (Martínez et al, 2019); moreover, mammalian nUTRs are enriched in Pum motifs, and interactions between nUTR genes and Pum proteins are conserved in mouse (Fig. 1), raising the possibility that the nUTR-dependent regulation of mRNA localization by Pum proteins is a conserved feature of neuronal development and function.

Complementing the notion that long 3′ UTRs specifically regulate localization and protein output of the mRNA that carries them, our study supports the existence of a "passive" mode of mRNA localization in neurons, in which nUTRs act as a regulatory entity: increased Pum binding to nUTR-containing isoforms helps contain transcripts in the somatic compartment. This goes in line with reports that short 3′ UTRs of tandem-UTR genes tend to be neurite-localized in mouse cortical and mESC-derived neurons (Ciolli Mattioli et al, 2019; Taliaferro et al, 2016), and does not contradict findings that 3′ UTR isoforms of localized (to either compartment) transcripts are significantly longer compared to those of non-localized transcripts, although neuropil regions of the rodent brain were reported to contain the longest 3′ UTR isoforms (Tushev et al, 2018). mRNAs with longer 3′ UTRs are generally less stable, presumably because they tend to harbor more destabilizing motifs such as AU-rich elements (AREs) (Mitschka and Mayr, 2022; Siegel et al, 2022). Indeed, the microRNA let-7 and ARE-binding proteins preferentially destabilize target mRNAs in the

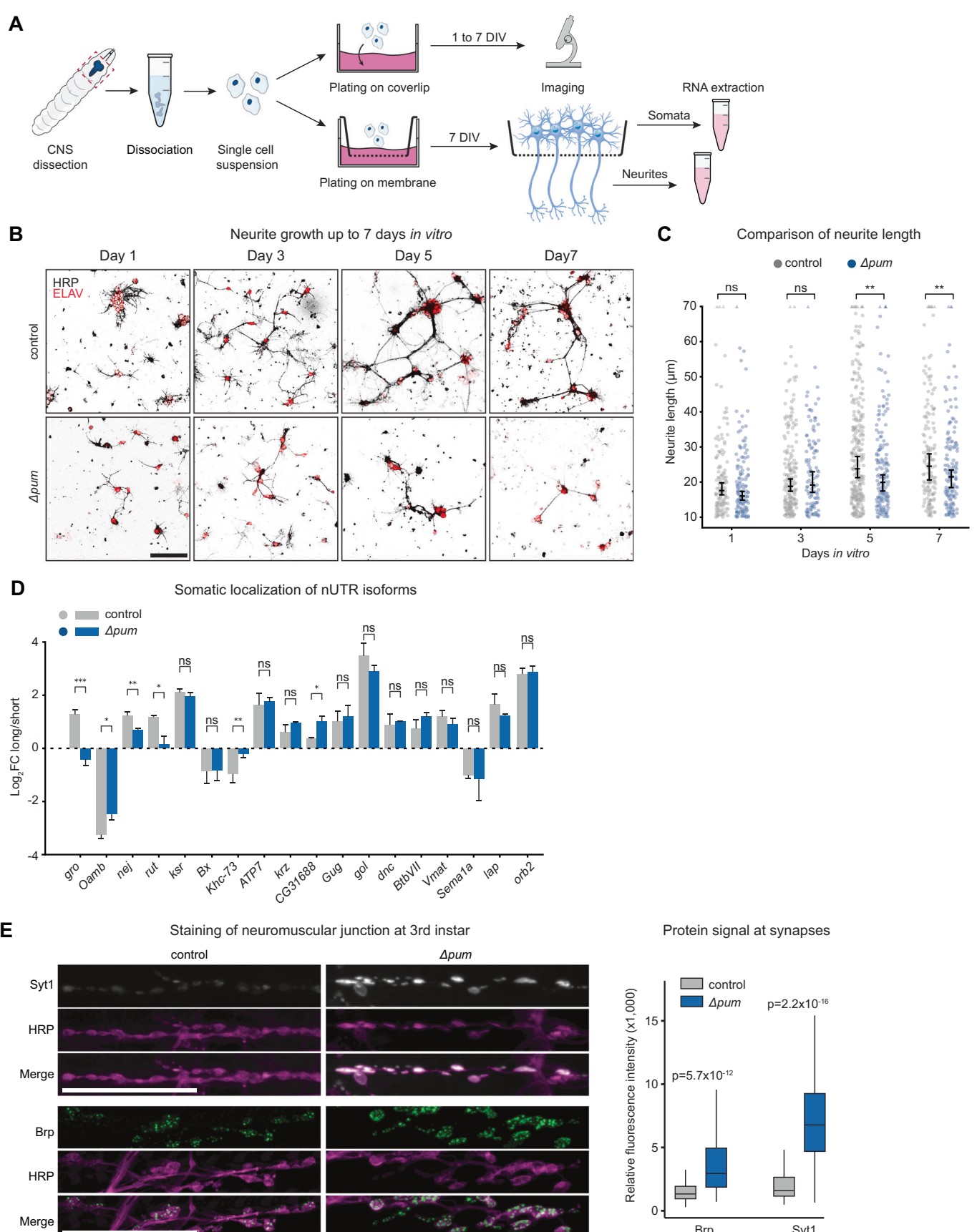

**Figure A:** CNS dissection → Dissociation → Single cell suspension → Plating on coverlip (1 to 7 DIV) → Imaging; Plating on membrane (7 DIV) → RNA extraction (Somata, Neurites)

**B** Neurite growth up to 7 days *in vitro*
HRP / ELAV
control / *Δpum*
Day 1 | Day 3 | Day 5 | Day7

**C** Comparison of neurite length
control / *Δpum*
Neurite length (μm) vs Days *in vitro* (1, 3, 5, 7)
ns, ns, **, **

**D** Somatic localization of nUTR isoforms
control / *Δpum*
Log$_2$FC long/short
gro, Oamb, nej, rut, ksr, Bx, Khc-73, ATP7, krz, CG31688, Gug, gol, dnc, BtbVII, Vmat, Sema1a, lap, orb2
***, *, **, *, ns, ns, **, ns, ns, *, ns, ns, ns, ns, ns, ns, ns

**E** Staining of neuromuscular junction at 3rd instar
control / *Δpum*
Syt1, HRP, Merge
Brp, HRP, Merge

Protein signal at synapses
control / *Δpum*
Relative fluorescence intensity (x1,000)
Brp p=5.7x10$^{-12}$, Syt1 p=2.2x10$^{-16}$

◄

**Figure 5.   Impaired neurite outgrowth, mRNA delocalization, and synaptic protein overexpression in neurons of *Δpum* flies.**

(A) Schematic of primary neuronal cell culture procedure. Nervous systems are dissected from third-instar *Drosophila* larvae and enzymatically and mechanically dissociated; the cells in the resulting suspension are allowed to divide and differentiate either on a coverslip for imaging, or on a microporous membrane for separation of neurite and soma compartments. DIV, days in vitro. (B) Confocal imaging of neurite phenotypes in cultured neurons of control (*w^1118*) and *Δpum* (*pum^ET7*/*pum^ET9*) flies, at the indicated days after plating. ELAV and HRP antibodies mark neuronal nuclei and neuronal cell membranes, respectively. Scale bars: 50 μm. (C) Quantification of neurite length in cultured neurons of control (*w^1118*) and *Δpum* (*pum^ET7*/*pum^ET9*) flies at the indicated days after plating. Neurites with a length >70 μm are represented as triangles at the top of the plot; neurites with a length <10 μm were excluded from the analysis. Error bars represent the mean ± SD of at least 119 neurites for each genotype and time point. **$p < 0.01$; p(5DIV control vs. 5DIV *Δpum*) = 1.071e−3, p(7DIV control vs. 7DIV *Δpum*) = 1.543e−3 (two-tailed Student's t-test). ns, not significant. Total number of neurites quantified $n = 1365$. (D) RT-qPCR quantification of the expression of nUTR-containing (long) relative to total (short) mRNA isoforms for the indicated genes, in isolated cell bodies of control (*w^1118*) and *Δpum* (*pum^ET7*/*pum^ET9*) flies, at seven days in vitro. Error bars represent the mean ± SD of three biological replicates for each genotype. ***$p < 0.001$, **$p < 0.01$, *$p < 0.05$; p(gro control vs. gro *Δpum*) = 2.166e−4, p(Oamb control vs. Oamb *Δpum*) = 1.299e−2, p(nej control vs. nej *Δpum*) = 1.611e−3, p(rut control vs. rut *Δpum*) = 1.975e−2, p(Khc-73 control vs. Khc-73 *Δpum*) = 7.861e−3, p(CG31688 control vs. CG31688 *Δpum*) = 1.603e−2 (two-tailed Student's t-test). (E) Confocal imaging (left) and quantification (right) of Syt1 and Brp expression in neuromuscular junctions of control (*w^1118*) and *Δpum* (*pum^ET7*/*pum^ET9*) third-instar larvae. HRP marks presynaptic membranes. Scale bars: 50 μm. Number of synapses scored: $n = 156$ (Syt1) and $n = 62$ (Brp) in control, $n = 178$ (Syt1) and $n = 88$ (Brp) in *Δpum*. Synapses with a diameter <4 μm were excluded from the analysis. *p*-values (two-tailed Student's t-test) are indicated. Boxes indicate range between minimum and maximum, the central line depicts the median, lower and upper bounds represent the first and third quartiles, respectively. Source data are available online for this figure.

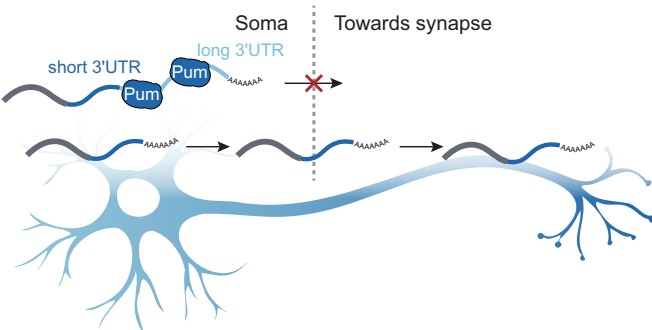

**Figure 6.   Model of 3′ UTR-dependent localization of Pum target mRNAs.**

In neuronal cell bodies, Pum binds to mRNAs encoding synaptic proteins, with a preference for long 3′ UTR isoforms. This leads to the enriched localization of translationally competent short isoforms in synaptic compartments.

soma of neurons, thereby causing an enrichment of mRNAs harboring let-7 motifs in neurites (Mendonsa et al, 2023). Overall, stable mRNAs and mRNA isoforms are enriched in neuronal projections and synaptic regions, where they are efficiently translated (Glock et al, 2021; Loedige et al, 2023; Tushev et al, 2018). Together, this body of evidence suggests that the nUTR-mediated destabilization and translational inhibition of long mRNA isoforms by factors such as let-7 and Pum in cell bodies, coupled to the escape of short 3′ UTR isoforms from this repression, accounts for the differential localization of many mRNA isoforms, leading to the compartment-specific function of the encoded protein (Fig. 6).

Binding of the long isoform in neuronal cell bodies by Pum may serve cellular functions beyond mRNA regulation. The relative enrichment of Pum binding motifs in cell bodies and distal compartments could help maintain precise levels of localized Pum protein. In flies and mammals, Pum plays important roles in apparently distinct molecular contexts, such as transcription and genome stability—in the nucleus—and synaptic function; accordingly, Pum localizes both in cell bodies as well as at distal compartments including neurites and neuromuscular junctions, often within discrete particles (Elguindy and Mendell, 2021; Menon et al, 2004; Vessey et al, 2006). It will be interesting to uncover the direct biological implications of these RNA/protein interactions.

## Methods

### Reagents and tools table

| Reagent/Resource | Reference or Source | Identifier or Catalog Number |
|---|---|---|
| **Experimental models** | | |
| *D. melanogaster:* *w^1118* | Bloomington stock center | BDSC:5905; RRID: BDSC_5905 |
| *D. melanogaster:* *pum^ET7* | Forbes et al (1998) | |
| *D. melanogaster:* *pum^ET9* | Forbes et al (1998) | |
| *D. melanogaster:* *pum^Flag* | This study | |
| **Recombinant DNA** | | |
| **Antibodies** | | |
| Mouse anti-CSP | DSHB | DCSP-2 (6D6) |
| Mouse anti-Synapsin | DSHB | 3C11 |
| Mouse anti-Syntaxin | DSHB | 8C3 |
| Mouse anti-Discs large | DSHB | 4D3 |
| Mouse anti-alphaTubulin | DSHB | AA4.3 |
| Rabbit anti-ELAV | Carrasco et al (2020) | |
| Mouse anti-LaminC | DSHB | LC28.26 |
| Rabbit anti-Histone H3 | Abcam | Ab1791 |
| Protein A | Life Technologies | 101023 |
| Mouse anti-Flag | Sigma | A8592 |
| Rat anti-ELAV | DSHB | 7E8A10 |
| Goat anti-HRP | Jackson ImmunoResearch | 123-605-021 |
| Donkey anti-rat | Thermo | 21208 |
| Mouse anti-Syt1 | DSHB | 3H2 2D7-s |
| Mouse anti-BRP | DSHB | nc82-c |

| Reagent/Resource | Reference or Source | Identifier or Catalog Number |
|---|---|---|
| Donkey anti-mouse | Abcam | ab150106 |
| Rabbit anti-HA | Cell Signaling Technology | 3724 |
| Goat anti-rabbit | Thermo Fisher Scientific | A-11008 |
| **Oligonucleotides and other sequence-based reagents** | | |
| sgRNA PumFlag | This study | ACGGCAACGTTGTGCTGTAA |
| PumFlag homology fragment | This study | Dataset EV5 |
| qPCR Primers | This study | Dataset EV5 |
| **Chemicals, Enzymes and other reagents** | | |
| Anti-Flag M2 magnetic beads | Invitrogen | M8823 |
| BSA | Sigma-Aldrich | A6947 |
| Concanavalin A | Sigma-Aldrich | C-2010 |
| DAPI | Abcam | AB228549 |
| Durcupan resin | Sigma-Aldrich | – |
| FastStart SYBR Green Master | Roche | 4913914001 |
| Fetal Bovine Serum | Gibco | A5256701 |
| Flag peptide | Sigma | F3290 |
| Insulin | Sigma-Aldrich | I6634 |
| iScript gDNA Clear cDNA Synthesis Kit | Bio-Rad | 1725034 |
| Laminin | VWR | 47743-734 |
| LDS sample buffer | FisherScientific | NP0007 |
| Liberase DH | Roche | 5401054001 |
| Lexogen QuantSeq 3′ mRNA-Seq Kit | Lexogen | – |
| Paraformaldehyde | FisherScientific | 28908 |
| Protease inhibitor cocktail | Roche | 11873580001 |
| Proteinase K | Ambion | AM2546 |
| RiboLock RNase inhibitor | Thermo Fischer Scientific | EO0384 |
| Schneider's insect medium | Gibco | 21720-024 |
| Triton X-100 | Sigma-Aldrich | X100 |
| TRIzol LS Reagent | Ambion | 10296028 |
| Tween20 | Sigma-Aldrich | P1379 |
| VECTASHIELD mounting medium | Vector Laboratories | H-1000 |
| **Software** | | |
| ImageJ Version 2.14.0/1.54f | Schindelin et al (2012) | https://imagej.nih.gov/ij/index.html |
| Rstudio Version 4.1.3. | R Development Core Team (2021) | http://www.rstudio.com/ |

| Reagent/Resource | Reference or Source | Identifier or Catalog Number |
|---|---|---|
| STAR v2.6.1b | Dobin et al 2013 | https://github.com/alexdobin/STAR |
| fastp | Chen et al (2018) | https://github.com/OpenGene/fastp. |
| DESeq2 | Love et al 2014 | https://bioconductor.org/packages/release/bioc/html/DESeq2.html |
| featureCounts | Liao et al (2014) | |
| MEME Suite 5.5.0 AME | McLeay and Bailey, 2010 | https://meme-suite.org/meme/tools/ame |
| HOMER | Heinz et al (2010) | |
| ggplot2_3.2.1 | | https://github.com/tidyverse/ggplot2 |
| dplyr_1.0.8 | | https://github.com/tidyverse/dplyr |
| MaxQuant (v. 1.6.14.0) | | https://www.maxquant.org/ |
| SNT | Arshadi et al (2021) | https://github.com/morphonets/SNT/tree/SNT-4.2.1 |
| **Other** | | |
| 13 mm plastic coverslips | Sarstedt | 831.840.002 |
| Anti-Flag M2 magnetic beads | Invitrogen | M8823 |
| BSA | Sigma-Aldrich | A6947 |
| Cell scraper | FisherScientific | 8100241 |
| Concanavalin A | Sigma-Aldrich | C-2010 |
| DAPI | Abcam | AB228549 |
| Durcupan resin | Sigma-Aldrich | – |
| FastStart SYBR Green Master | Roche | 4913914001 |
| Fetal Bovine Serum | Gibco | A5256701 |
| Flag peptide | Sigma | F3290 |
| Insulin | Sigma-Aldrich | I6634 |
| iScript gDNA Clear cDNA Synthesis Kit | Bio-Rad | 1725034 |
| KONTES Tissue Grinder | VWR | – |
| Laminin | VWR | 47743-734 |
| LDS sample buffer | FisherScientific | NP0007 |
| Liberase DH | Roche | 5401054001 |
| Lexogen QuantSeq 3′ mRNA-Seq Kit | Lexogen | – |
| Illumina Stranded mRNA Prep | Illumina | 20040534 |
| Microporous membrane inserts (1 μm) | cellQART | 931 10 12 |
| NovaSeq6000 platform | Illumina | |
| CM100 microscope | Philips | |

| Reagent/Resource | Reference or Source | Identifier or Catalog Number |
|---|---|---|
| Orbitrap | FisherScientific | |
| EASY-nLC 1200 | FisherScientific | |
| 2100 Bioanalyzer | Agilent Technologies | |

## Drosophila melanogaster husbandry and mutagenesis

Experiments in this study used larvae or adult *Drosophila melanogaster* flies, males and females (in equal amounts). Flies were raised at 25 °C. $w^{1118}$ flies were obtained from the Bloomington stock center (5905). The *pumilio* loss-of-function alleles $pum^{ET7}$ and $pum^{ET9}$ were obtained from Ruth Lehmann (Forbes and Lehmann, 1998) and kept in heterozygosis over GFP-marked balancer chromosomes. To generate $pum^{Flag}$ flies, which express an endogenously, C-terminally Flag-HA-tagged Pum protein, a guide RNA (acggcaacgttgtgctgtaa) targeted the *pum* locus at chromosome 3. Genome editing (Port and Bullock, 2016) used a 2602 bp homology donor fragment (sequence listed in Dataset EV5). Embryo injection was performed by Bestgene, Inc.

## Drosophila primary neuron cultures

In vitro cultures of Drosophila larval neurons were performed as described in Smrt et al (Smrt et al, 2015), with modifications. Third-instar larvae were cleaned in 95% EtOH, rinsed three times in water and central nervous systems (CNSs) were dissected. 3 larval CNSs per sample were incubated in enzyme solution (0.21 Wünsch units/mL Liberase DH (Roche 5401054001) in Rinaldini's saline) for 1 h at room temperature (RT). CNSs were gently washed twice with 1 mL culture medium (Schneider's insect medium (Gibco 21720-024) supplemented with 10% fetal bovine serum (FBS) (Gibco A5256701), 50 µg/mL insulin (Sigma-Aldrich I6634). Cells were dissociated in 200 µL culture medium by pipetting CNSs up-and-down against the bottom of the tube 100 times. The cells were plated on 13 mm plastic coverslips (Sarstedt no. 83.1840.002) or microporous membrane inserts (cellQART 931 10 12, pore size 1 µm) coated with Concanavalin A (Sigma-Aldrich C-2010) and Laminin (VWR 47743-734). Coverslips were coated on top, cell culture inserts were coated only on the bottom. Cells were allowed to adhere for 2 h at 25 °C, 80% humidity. Culture medium containing cell debris and non-adherent cells was removed and cells attached to the coverslip were flooded with 500 µL culture medium. Cells were allowed to grow in 25 °C, 80% humidity for up to 7 days.

## Soma/neurite separation in Drosophila primary neuron cultures

Three *Drosophila* 3rd instar larval brains were dissected for each sample. Primary neurons were incubated for seven days in vitro. After aspiration of the medium, the cell culture inserts (cellQART 931 10 12) were transferred to a new 12-well plate containing 1 mL of ice-cold PBS. The cell culture insert was inverted and neurites were removed using a cell scraper (FisherScientific 08100241) pre-

wetted in 250 µL ice-cold PBS. The neurites were transferred by washing the cell scraper in 250 µL ice-cold PBS. After the neurite removal, the cell culture insert was inverted again and the somata were removed by adding 250 µL ice-cold PBS to the insert and pipetting up/down 25 times. 750 µL TRIzol LS Reagent (Ambion 10296028) were added immediately to each 250 µL- sample and RNA was extracted according to the manufacturer's instructions.

## Immunofluorescence staining and analysis of cultured neurons

Six *Drosophila* 3rd instar larval brains were dissected per genotype, resulting in 1.5 larval brains used per coverslip and time point. Cells were fixed at RT for 10 min in 4% paraformaldehyde (PFA) (FisherScientific 28908) in PBS, washed twice for 5 min in PBS and stored in PBS at 4 °C. Cells were blocked for 1 h at RT with 3% BSA (Sigma #A6947 LOT #SLBZ6761) in 0.2% PBS-Triton X-100 (Sigma-Aldrich #X100). Primary antibodies were incubated overnight at 4 °C in 0.2% PBS-Triton X-100, rinsed once with PBS and washed thrice 5 min with PBS. The secondary and conjugated antibodies were incubated for 1 h at RT in the dark, rinsed once with PBS and washed thrice 5 min with PBS. The penultimate wash included a counterstaining with 4′,6-diamidino-2-phenylindole (DAPI) (Abcam #AB228549) for 10 min at RT. The coverslips were mounted using VECTASHIELD antifade mounting medium (Vector Laboratories #H-1000) and imaged using a Zeiss LSM 900 confocal microscope in a sequential scanning mode. Neurites were traced in a semi-automated manner using the SNT framework for neuronal morphometry (Arshadi et al, 2021) on the Fiji (ImageJ) platform. To trace a neurite, the start point was set at the edge of the cell soma; a subsequent point on the neurite was selected next and the neurite was traced automatically between the two points.

## Dissection of neuromuscular junction and immunofluorescence staining

Third-instar larvae were collected and rinsed in 1xPBS to remove debris. Neuromuscular junction (NMJ) dissection followed the protocol by Brent, Werner and McCabe (Brent et al, 2009). Peripheral body walls ("larval filets") were fixed using 4% PFA in 1xTBS for 15 min at RT. Filets were washed for 5 min at RT in 1% TBS-Tween20 (Sigma-Aldrich #P1379), blocked for 3 h at 38 °C in 5% BSA (Sigma #A6947 (LOT #SLBZ6761)) in 0.5% TBS-Triton X100 (Sigma-Aldrich #X100) on a shaker. Primary antibodies were incubated for 10 min at 38 °C and then 3 days at 4 °C in 5% BSA in 0.1% TBS-Triton X100. Following three washes for 15 min at RT with 1xTBS, secondary antibodies were incubated for 4 h at 35 °C in 5% BSA in 0.1% TBS-Triton X100 in the dark. Filets were washed three times for 15 min at RT with 1xTBS, mounted using VECTASHIELD antifade mounting medium (Vector Laboratories #H-1000) and imaged using a Zeiss LSM 900 confocal microscope in Z-project scanning mode.

## Quantification of synaptic SYT1 and BRP expression

Images were analyzed using the Fiji (ImageJ) platform. Z-stacks were imported, the recorded channels split and a maximum intensity projection created. Using the ROI Manager, synapses were marked and the selected areas were copied to the channel of interest

(Syt1 or Brp). To not introduce any bias, the images were randomly numbered and the synapses were marked in the HRP channel. Mean intensities and sizes of the marked areas were exported and quantified.

## Antibodies and protein detection

For western blots, mouse anti-CSP (DSHB #DCSP-2 (6D6)), mouse anti-Synapsin (DSHB #3C11), mouse anti-Syntaxin (DSHB Cat#8C3), mouse anti-Discs large (DSHB Cat#4D3), mouse anti-alphaTubulin (DSHB Cat#AA4.3), rabbit anti-ELAV (Carrasco et al, 2020), mouse anti-LaminC (DSHB Cat#LC28.26) and rabbit anti-Histone H3 (Abcam #Ab1791) were used at concentrations 1:150, 1:400, 1:250, 1:350, 1:100, 1:1000, 1:450, and 1:10,000, respectively. Peroxidase-conjugated antibodies were used at the following concentrations: protein A (Life Technologies #101023) 1:4000, anti-Flag (Sigma #A8592) 1:10,000. For immunofluorescence of cultured primary neurons, rat anti-ELAV (DSHB #7E8A10), goat anti-HRP (Jackson ImmunoResearch #123-605-021), and donkey anti-rat (Thermo #21208) were used at concentrations: 1:50, 1:50, and 1:750, respectively. For NMJ immunofluorescence, mouse anti-Syt1 (DSHB #3H2 2D7-s), mouse anti-BRP (DSHB #nc82-c), goat anti-HRP (Jackson ImmunoResearch #123-605-021) and donkey anti-mouse (Abcam #ab150106) were used at concentrations: 1:50, 1:50, 1:50, and 1:500, respectively. For immunofluorescence of primary neurons of *pum^Flag^* flies, in which Pum is Flag-HA tagged, rabbit anti-HA (Cell Signaling Technology #3724) and goat anti-rabbit (Thermo Fisher Scientific # A-11008) were used at concentrations of 1:200 and 1:750, respectively.

## Pumilio cross-linking RNA immunopurification (Pum xRIP)

400 mg UV-cross-linked head powder ($6 \times 300$ mJ/cm²) from either *pum^Flag^* or *w^1118^* adult flies was homogenized in 4 mL lysis buffer (200 mM NaCl, 50 mM HEPES-KOH pH 7.9, 1 mM EDTA, 0.5 mM EGTA) supplemented with 0.5 mM DTT and 0.2% TritonX. Unless specified otherwise, all buffers were supplemented with a protease inhibitor cocktail (Roche 11873580001) and RiboLock RNase inhibitor (Thermo Fischer Scientific EO0384). After homogenization, detergents were added to a final concentration of 0.5% SDS and 0.5% Na-deoxycholate and the samples were incubated 5 min on ice. The homogenate was centrifuged twice at $15,000 \times g$ for 10 min at 4 °C, each time transferring the supernatant to a new tube. Ionic detergents were quenched with 1% NP-40. One xRIP sample consists of 750 µL processed lysate (input) incubated with 40 µL anti-Flag M2 magnetic beads (Invitrogen M8823) for 1.5 h at 4 °C. Beads were rinsed with lysis buffer containing 0.1% SDS and 0.1% Na-deoxycholate, washed three times for 5 min at 4 °C with lysis buffer containing 0.1% SDS, 0.1% Na-deoxycholate and 1% TritonX, and twice with lithium chloride wash buffer (350 mM LiCl, 50 mM HEPES-KOH pH 7.5, 1 mM EDTA, 1% NP-40, 0.7% Na-deoxycholate) for 5 min at 4 °C. Cross-linked protein–RNA complexes were eluted in 120 µL elution buffer (10 mM Tris-HCl pH 7.4, 150 mM NaCl, 0.1% TritonX, no protease inhibitor) containing 0.2 mg/mL Flag peptide (Sigma F3290), for 1 h at 4 °C. Eluates and corresponding inputs were supplemented with 10x Proteinase K buffer (10 mM Tris-HCl pH 7.5, 100 mM NaCl,

10 mM EDTA, 0.5% SDS, no protease inhibitors) subjected to protein digestion with 30 µg of Proteinase K (Ambion AM2546) for 45 min at 50 °C and 1100 rpm. RNA was purified with TRIzol LS Reagent (Ambion 10296028) according to the manufacturer's instructions.

## Isolation of Drosophila synaptosomes

~25 g of anesthetized 3-day-old *w^1118^* flies were collected in a kitchen blender (Philips ProBlend 6) in 400 mL ice-cold PBS and blended with 5 short pulses. The homogenate was poured through a sieve system (grid size top-to-bottom: 710 µm - 425 µm - 355 µm - 125 µm). Each sieve was washed with pressurized water to separate different fly parts. Heads were collected in 355 µm sieves and transferred into a tube containing sucrose buffer (0.32 M sucrose, 1 mM EDTA, 5 mM Tris pH 7.4, 0.25 M DTT, supplemented with 1x Protease inhibitor cocktail Roche, 11873580001 and 1x RiboLock RNase inhibitor Thermo Fisher Scientific, EO0384) with a spatula. The suspension containing heads was poured through a 100 µm cell strainer to separate tissue from the collection sucrose buffer and separated into 4 equally aliquots that were each separately homogenized with 7 gentle strokes in a KONTES Tissue Grinder (VWR) (loose pestle) in 10 mL sucrose buffer. Homogenates were pooled and poured through a 40 µm strainer. The input sample for RNA sequencing and proteomics was collected at this step. The homogenate was centrifuged twice at $1000 \times g$ for 20 min at 4 °C, then at $2000 \times g$ for 20 min at 4 °C, each time carefully transferring the supernatant to a new tube, and finally at $15,000 \times g$ for 15 min at 4 °C. The resulting pellet was resuspended in 2 mL sucrose buffer, loaded onto a precooled Percoll gradient (from top to bottom 3, 10, 15, 23% Percoll diluted in sucrose buffer (0.32 M sucrose, 1 mM EDTA, 5 mM Tris pH 7.4, 0.25 mM DTT, 1x Protease inhibitor cocktail and 1x RiboLock RNase inhibitor) and centrifuged at $31,000 \times g$ at 4 °C for 5 min (acceleration 9, deceleration 7). Resulting layers contained membranes (fraction 1 and 2), membranes and synaptosomes (fraction 3), synaptosomes (fraction 4) and mitochondria (fraction 5) (see also Fig. 2). Synaptosome fractions (fractions 3 and 4) were diluted in 30 mL sucrose buffer and centrifuged at $20,000 \times g$ for 35 min 4 °C (acceleration 9, deceleration 6). Most of the supernatant was discarded, leaving ~65 µl of volume atop to not disturb the pellet containing synaptosomes. Each synaptosome sample was split in two: one half was diluted in TRIzol LS Reagent (Ambion, 10296028) for RNA purification according to the manufacturer's instructions. The other half was diluted in RIPA buffer (50 mM tris-HCl pH 8, 150 mM NaCl, 1% NP-40, 0.5% Na-deoxycholate, 0.1% SDS, 5 mM TCEP), homogenized for 1 min with a mechanical Pellet Pestle Motor (KONTES), incubated for 15 min on ice, centrifuged 5 min at $16,000 \times g$ at 4 °C to clear the lysate, and submitted for proteomics analysis.

## Mass spectrometry on Drosophila synaptosomes

Synaptosome and input fractions derived from three biological replicates were compared. Corresponding samples (10 µg total protein in RIPA buffer containing 5 mM TCEP) were adjusted to 1xLDS sample buffer (Thermo Fisher NP0007; 5 mM TCEP final concentration) and heated for 5 min at 95 °C. Proteins were separated on a 4–12% Novex BOLT gel (Thermo Fisher) and

stained with colloidal Coomassie (InstantBlue, Expedeon). For each replicate, entire gel lanes were cut into five evenly distant slices, which were further processed by standard trypsin (Promega) in-gel digestion procedure. Following peptide clean-up by C18-STAGE tipping, tryptic peptides were analyzed by nanoLC-MS (Thermo Fisher nLC1200 hyphenated to a Q Exactive Plus MS) using a one column liquid-junction setup, in which the in-house packed (Dr. Maisch, Reprosil-Pur C18 AQ 120A 1.9 μm beads) analytical capillary column (NewObjective, 300 mm L, 360 μm OD, 75 μm ID, 8 μm tapered open end) concomitantly served as the ESI (electrospray ionization) emitter. A "120 min" nLC-MS method was conducted via single injection measurements (one technical replicate). The gradient (HPLC solvent A: 0.1% formic acid; solvent B: 0.1% formic acid, 80% acetonitrile) for the "120 min" method was: 0 min: 2%, 5 min: 4%, 90 min: 30%, 10 min: 40% 2 min: 80% (%B buffer; 300 nl/min flow rate). The wash out step was as follows. 5 min: 80%, a 2 min inverse gradient from 80% to 2% and a 5 min re-equilibration step at 2% B buffer (flow rates 500 nl/min). All measurements were carried out in data-dependent (DDA) mode employing a Top12 (Q Exactive Plus) ion selection/fragmentation regimen.

## Mass spectrometry analysis on Q Exactive Plus

Survey full scan MS spectra ($m/z$ 300–1650) were acquired in the Orbitrap with 70,000 resolution ($m/z$ 200) after accumulation of ions at a normalized AGC target of $3 \times 10^6$ at an expected LC peak width of 20 s and a default charge state of 2. Dynamic exclusion was set to 20 s (mass tolerance of ±10 ppm). The 12 most intense multiply charged ions ($z \geq 2 \leq 6$) were sequentially isolated (2.0 amu window) and fragmented in the octopole by higher-energy collisional dissociation (HCD) with a fixed maximum injection time of 60 ms at a normalized AGC target value of $1 \times 10^5$ and 17,500 resolution. General mass spectrometric conditions were as follows. Spray voltage, 2.1 kV; heated capillary temperature, 275 °C; normalized HCD collision energy 28%, RF lens value 60% and MS/MS ion minimal AGC target value set to $1.2 \times 10^4$ counts.

## Mass spectrometry data analysis

MaxQuant (v. 1.6.14.0) employing standard parameters (enabling match between runs in between groups, match window 0.5 min) was used to identify peptides and final protein identification role-up (both at 1%FDR). MS raw data were searched simultaneously with the target-decoy standard settings against the Uniprot *Drosophila melanogaster* database (Uniprot_reviewed+Trembl including canonical isoforms; downloaded on August 2020) concatenated with an in-house curated FASTA file containing commonly observed contaminant proteins. Cysteine carbamido-methylation (+57.021464 Da) was set as a fixed modification and N-acetylation of protein (N-term +42.010565 Da), NQ deamida-tion (+0.984016 Da) and methionine oxidation (+15.994915 Da) as variable modifications. Maximum (allowed) variable PTMs per peptide were set to four. The MaxLFQ algorithm was utilized for relative label-free quantification (min. ratio count ≥1) together with the iBAQ information. The MaxQuant output (proteingroups. txt; $\log_2$ transformed LFQ values) was further analyzed by standard R packages (removal of contaminant and reverse database hit IDs, vsn normalization, missing value imputation, limma package

moderated T test and multiple testing correction). The fold change of protein abundance in synaptosomes vs. input was determined by calculating the difference between the Log2 transformed LFQ intensity values in the respective fractions. Proteins with Log2FC > 0.6 (after rounding up to the 2nd decimal place) and *p*-value < 0.05 (moderated limma *p*-value; not adjusted) were classified as enriched in synaptosomes.

## Transmission electron microscopy on purified synaptosomes

Synaptosomes were fixed in a solution containing 4% paraformal-dehyde (PFA, w/v) and 2% glutaraldehyde (w/v) in phosphate buffer (PB, 0.1 M, pH 7.4) for 30 min at RT. After fixation, the synaptosomes were washed with PB and incubated with 1% osmium tetroxide for 30 min. After every step, the synaptosomes were centrifuged at $7000 \times g$ for 3 min. Before dehydration, the pellet was coated with Agar. Graded ethanol (up to 50%) washes were performed for 5 min each, followed by incubation in 1% uranyl acetate for 30 min. The synaptosomes were then dehydrated with graded ethanol solutions (80%, 90%, 98% once for 5 min each, 100% twice for 10 min) and incubated in 100% propylene oxide (twice for 10 min) and for 1 h in Propylenoxid/Durcupan 1:1. Subsequently, the synaptosomes were embedded in Durcupan resin (Sigma-Aldrich). Ultrathin sectioning (55 nm) was done with a Leica UC6 Ultracut. Sections were mounted onto copper grids (Plano) and contrasted using Pb-citrate for 3 min. Electron microscopy was performed using a Philips CM100 microscope equipped with a Gatan Kamera Orius SC600 at magnifications from 700× to 2950×. Image analysis was carried out with ImageJ/Fiji.

## RT-qPCR

For synaptosome RT-qPCR, 500 ng total RNA was used for each sample. Reverse transcription used iScript gDNA Clear cDNA Synthesis Kit (Bio-Rad). RT-qPCR was performed in a LightCycler 480 II instrument using FastStart SYBR Green Master (Roche 04913914001). For the RT-qPCR of neurite and soma separation, 200 ng total RNA was used for each sample. Results of the soma and neurite fractions, which originated from the same biological replicate, were analyzed pairwise. Three biological replicates were used. RT-qPCR primer sequences are listed in Dataset EV5.

## mRNA sequencing library preparation

RNA quality was tested using a 2100 Bioanalyzer (Agilent Technologies). Libraries for mRNA sequencing were prepared using Illumina Stranded mRNA Prep (Illumina 20040534) and paired-end sequencing was performed using the NovaSeq6000 platform (Illumina) and 150 bp reads.

## 3′-end sequencing (3′-seq)

10 ng of total RNA from input or Pum xRIP, or 15 ng of total RNA from each purified fraction (input and synaptosome) was used for library preparation using Lexogen QuantSeq 3′ mRNA-Seq reverse library prep kit (Lexogen) according to manufacturer's instructions. Paired-end sequencing was performed using the NovaSeq6000 platform (Illumina) and 2 × 150-bp reads.

## 3′-end sequencing data analysis

Two 3′-seq datasets were analyzed: a Pum xRIP-3′-seq experiment to assess Pum target mRNAs, and a synaptosome 3′-seq experiment to assess synaptic mRNA isoforms. Reads of both datasets were trimmed with fastp (Chen et al, 2018) and mapped with STAR (v2.6.1b; (Dobin et al, 2013)) to release 93 of the dm6 genome (parameter settings: --sjdbOverhang 74 --limitBAMsortRAM 60000000000 --alignIntronMax 1; all others default). Unless stated otherwise, Pum xRIP-3′-seq analyses were performed on the dataset 'Run1' in the deposited record GSE233390.

## 3′-end site definition and filtering

For Pum xRIP-3′-seq, signal originating from internal priming was mitigated by blacklisting bases with downstream genomic stretches of As ≥6 bases long, or with 7 out of 10 downstream bases being As. Regions within 25 bases of Ensembl-annotated 3′-ends (Ensembl assembly release dm6) were not subject to blackout. Only 3′-seq signal within 500 bp downstream of an Ensembl-annotated gene was considered. In Synaptosome 3′-seq, internal priming signal was mitigated by blacklisting bases with downstream genomic stretches of As ≥4 bases long, or with 6 of 8 downstream bases being As. Regions within 25 bases of Ensembl 3′-ends were not subject to blackout. Only 3′-seq signal within 2000 bases downstream of an Ensembl annotated gene was considered. 3′-seq peaks were clustered outward up to 15 bases from peaks with a read depth ≥6 in a single sample. Intersecting clusters were merged into single clusters. Reads were then re-counted for all samples across these clusters. Both xRIP-3′-seq and Synaptosome 3′-seq clusters were further subjected to validation against a set of high-confidence 3′-ends as described in Alfonso-Gonzalez et al (Alfonso-Gonzalez et al, 2023). Briefly, these 3′-ends were either identified through direct-RNA long-read sequencing, or represent Ensembl (release 93) annotated 3′-ends that either (i) are the most distal 3′-end of a gene with no observed long-read derived 3′-end, or (ii) are a most-distal Ensembl 3′-end that is more distal than any observed long-read 3′-end. Only clusters intersecting 50-nt windows around these validated 3′-ends were considered in downstream length enrichment analyses and in differential gene/cluster expression analysis of the synaptosome 3′-seq dataset. Relative xRIP-3′-seq expression across proximal and distal (nUTR) 3′ UTR regions of 271 validated nUTR-containing genes, as described in Carrasco et al (Carrasco et al, 2020), was assessed as follows. xRIP-3′-seq signal within 100 bases of proximal and distal 3′-ends for each gene was whitelisted for respective regions. Additional signal within proximal and distal regions upstream of 100-base 3′-end windows was included as valid signal only if they intersected 50-nt windows around high-confidence 3′-ends identified and described in Alfonso-Gonzalez et al (Alfonso-Gonzalez et al, 2023). Expression of proximal and distal regions were then assessed in DESeq2 (Love et al, 2014).

## 3′ UTR length analysis in synaptosome 3′-seq

Normalized read counts of validated 3′-ends were exported from DESeq for each replicate and averaged for both conditions (Synaptosome and Input). 3′ UTR lengths of each 3′ UTR isoform were assigned by subtracting the most proximal 3′ UTR start site in the Ensembl dm6 (release 93) gene annotation from the 3′-end coordinate of each isoform. For genes showing more than one expressed 3′-end isoform (APA genes), the relative 3′-end isoform abundance was calculated for each of these isoforms as the percentage of total expression. To determine the expression-weighted 3′ UTR length of each APA gene in input and synaptosomes, the distal-most 3′ UTR length of each gene was normalized to 1, and the average 3′ UTR length was calculated by summing [% of distal length * % of total gene expression] of all isoforms of a gene. Alternatively, ranked length metric was enforced whereby the distance between 3′-ends of 3′ UTR isoforms was normalized to a ranked fraction of 1, with 1 representing the length of the most distal isoform (e.g., 0.33, 0.67, and 1 for a three-isoform gene). Average percent of ranked distal 3′ UTR length was calculated for each APA gene for both synaptosome and input. Example: a two-isoform gene with each isoform expressed equally (50% each), ranked 3′ UTR length = 0.5*50% proximal expressed + 1.0*50% distal Isoform expressed. Both $\log_2$-fold difference and subtracted difference between the average percent of 3′ UTR length in synaptosome vs. input were assessed using both the real length and rank-length metrics. 5% Z-score thresholds ($p < 0.05$, 1-tailed) were calculated on all four distributions. Any gene with $p < 0.05$ in either of these four distributions qualified that gene as displaying a significant 3′ UTR length change between input and synaptosome.

## Pum binding to long compared to short 3′ UTR isoform (Pum xRIP)

These Pum xRIP-3′Seq analyses were performed on the dataset 'Run2' in the deposited record GSE287595. Pum xRIP-3′Seq data were processed as described above. The analyzed datasets include input and xRIP samples for $pum^{Flag}$ flies and $w^{1118}$ flies. Cluster counts files per replicate were merged into a single counts object. Counts per condition were averaged from all the replicates. Clusters were then merged for each condition and gene based on them being proximal or distal to the gene-specific proximal poly(A) site. Proximal clusters were used to calculate the level of binding to the short 3′ UTR isoform and distal clusters for level of binding to the long 3′ UTR isoform.

## 3′ UTR isoform distribution analysis

43 genes whose short isoforms were enriched in the synapse fraction, AND bound by Pum, were selected for analysis. Regions for short and 3′ UTR isoform regions were defined as described above. Reads falling into the entire UTR region, which originate from both short and long 3′ UTR isoforms, and reads falling into regions distal to the proximal poly(A) site, which originate only from long isoforms, were counted using featureCounts (Liao et al, 2014) in the Subread package. The proportion of the long 3′ UTR isoform was calculated by dividing the distal counts by the total 3′ UTR read count.

## Motif analysis

Motifs of known *Drosophila melanogaster* RBPs as defined in Ray et al (Ray et al, 2013) were scanned for enrichment using the AME tool within the MEME motif analysis suite (v 5.5.0) (McLeay and Bailey, 2010). Using default AME parameters, the most distal expressed 3′ UTRs of 460 identified Pum target genes, and a subset of 71 nUTR-containing Pum target genes, were assessed against

backgrounds of the most distal expressed 3′ UTRs of all 8931 expressed genes, and 3757 expressed APA genes, respectively. The −max enrichment scoring was used in addition to default −avg, when scoring enrichment of motifs in 3′ UTRs of mRNAs encoding 989 synaptosome-enriched proteins vs. 3′ UTRs of a background of 4843 total genes detected by proteomics. HOMER motif analysis (Heinz et al, 2010) was used (find options: -offset 1 -strand +) to scan for presence and location of Pum motifs within proximal and nUTR regions of the set of nUTR-containing Pum target genes, and the 200 nUTR-containing non-target genes.

## microRNA enrichment analysis

microRNA target sites as defined in the TargetScanFly repository (v7.2) (Agarwal et al, 2018) were scanned in most distal expressed 3′ UTRs of mRNAs encoding 989 proteins enriched in the synaptosome fraction, and assessed for enrichment against a 3′ UTR background of 4843 proteins found in synaptosome or input. If a gene's 3′ UTR contained a microRNA target site, the gene was qualified as a microRNA target gene. Enrichment of 91 conserved microRNAs was assessed. $Log_2$-fold enrichment in the number of genes containing microRNA target sites was compared both with and without adjusting for the greater average 3′ UTR length of mRNAs encoding proteins enriched in synaptosome relative to the all-protein background. A parallel scan and enrichment assessment were carried out on a set of 211 genes with RNA isoforms enriched in synaptosomes against an all expressed (based on mRNA 3′-seq) gene background of 9034 genes.

## Gene Ontology analysis

GO enrichment analysis was performed using the DAVID online server (version 2021). Pum mRNA target genes (either all targets or targets with neuronal 3′ UTRs) or all genes with neuronal 3′ UTRs were queried against the background of RNAs expressed in Pum xRIP-3′-seq with base mean >10. Proteins enriched in the synaptosome fraction were queried against the background of all proteins detected in the synaptosome experiment. GO terms with Benjamini-Hochberg adjusted $p$-values < 0.01 (Fig. EV2E) or <0.05 (all other figures) were classified as significant.

## Data availability

All sequencing data generated during this study (RNA-Seq data, Pum xRIP-3′-seq Run1; RNA-Seq data, Pum xRIP-3′-seq; mRNA-seq data, cultured neurons) can be accessed at NCBI Gene Expression Omnibus under the accession number GSE233392. Pum xRIP-3′-seq Run2 can be accessed at NCBI Gene Expression Omnibus under the accession number GSE287595.

The source data of this paper are collected in the following database record: biostudies:S-SCDT-10_1038-S44319-025-00401-z.

## Peer review information

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

## Acknowledgements

We thank Fernando Mateos and Sarah Holec for technical help. We are grateful to Alejandro Gomez Auli and Gerhard Mittler at the Proteomics Core, Ulrike Bönisch and the Deep Sequencing Core, and Thomas Manke and the Bioinformatics Core at MPI-IE. We thank Dominic Grün, Andrew Straw, Olivier Urwyler and Dierk Reiff for helpful discussions, ideas and feedback. Stocks obtained from the Bloomington Drosophila Stock Center (NIH P40OD018537) were used in this study. This work was funded by the Max Planck Society, the Deutsche Forschungsgemeinschaft (DFG, German Research Foundation) SFB 1381 (Project-ID 403222702) and under Germany's Excellence Strategy (CIBSS – EXC-2189 – Project-ID 390939984), and the European Research Council (ERC) under the European Union's Horizon 2020 research and innovation program (grant agreement ERC-2018-STG-803258).

## Author contributions

**Dominika Grzejda**: Formal analysis; Validation; Investigation; Visualization; Methodology; Writing—original draft. **Anton Hess**: Formal analysis; Investigation; Visualization; Methodology; Writing—review and editing. **Andrew Rezansoff**: Formal analysis; Investigation. **Sakshi Gorey**: Formal analysis; Investigation. **Judit Carrasco**: Investigation. **Carlos Alfonso-Gonzalez**: Formal analysis; Investigation. **Stylianos Tsagkris**: Investigation. **Lena Neuhaus**: Investigation. **Mengjin Shi**: Formal analysis; Investigation; Visualization. **Hasan Can Ozbulut**: Formal analysis. **Friederike-Nora Vögtle**: Formal analysis; Investigation. **Andreas Vlachos**: Formal analysis; Investigation; Methodology. **Valérie Hilgers**: Conceptualization; Formal analysis; Supervision; Funding acquisition; Investigation; Visualization; Writing—original draft; Project administration; Writing—review and editing.

Source data underlying figure panels in this paper may have individual authorship assigned. Where available, figure panel/source data authorship is listed in the following database record: biostudies:S-SCDT-10_1038-S44319-025-00401-z.

## Funding

## Disclosure and competing interests statement

The authors declare no competing interests.

# Expanded View Figures

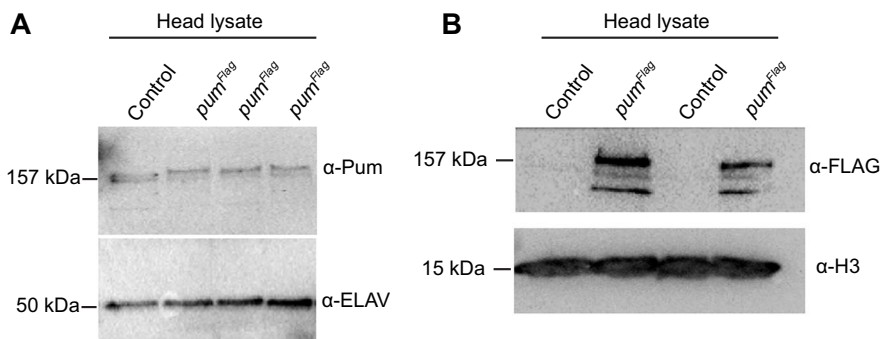

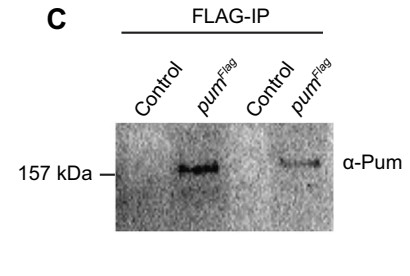

**A** Head lysate

157 kDa — α-Pum

50 kDa — α-ELAV

**B** Head lysate

157 kDa — α-FLAG

15 kDa — α-H3

**C** FLAG-IP

157 kDa — α-Pum

**D**

| Motif | RNA-binding protein | E-value | % occurrence in Pum targets | % occurrence in background |
|---|---|---|---|---|
| UGUA_A | PUM | 1.14e-43 | 74.1 | 38.0 |
| UUU Ggg | ORB2 | 1.29e-20 | 79.0 | 53.8 |
| GAAAA | PABP | 2.26e-19 | 71.7 | 46.5 |
| UU_ gUU | RBP9 | 1.38e-17 | 82.8 | 60.1 |
| UU_ gUU | FNE | 2.57e-16 | 75.2 | 52.0 |

**E**

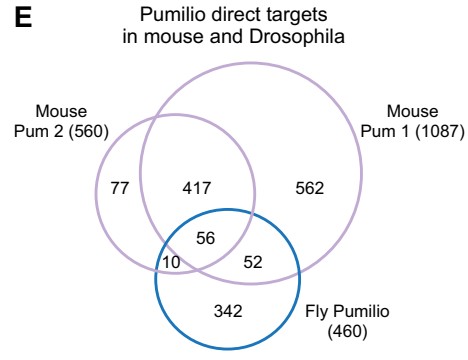

Pumilio direct targets in mouse and Drosophila

Mouse Pum 2 (560)

Mouse Pum 1 (1087)

77 | 417 | 562

56

10 | 52

342

Fly Pumilio (460)

**Figure EV1.  xRIP-3′-seq identifies mRNAs directly bound by Drosophila Pumilio.**

(A–C) Western blots showing Pum expression in adult fly heads. (A) Detection with an anti-Pum antibody shows the molecular weight difference between wild-type Pum (untagged) compared to Flag-HA-tagged Pum protein in three independent *pum^Flag* transformant flies. ELAV serves as loading control. (B) Detection with an anti-Flag antibody shows specific detection of Pum-Flag in *pum^Flag* flies. Histone H3 serves as a loading control. Five fly heads were used for protein preparation for each genotype. (C) Eluates from an anti-Flag antibody immunopurification of Pum from head extract of *pum^Flag* and control (*w^1118*) flies. (D) The Pumilio Response Element (PRE) constitutes the most enriched motif in the 3′ UTRs of Pum target mRNAs ($p = 1.14*10^{-43}$; E-value). Motifs for the top five RBPs are shown. (E) Venn diagram showing the intersection between Pum mRNA targets in Drosophila (this study) and for mammalian Pum 1 and Pum 2 identified in Zhang et al, 2017 (Zhang et al, 2017).

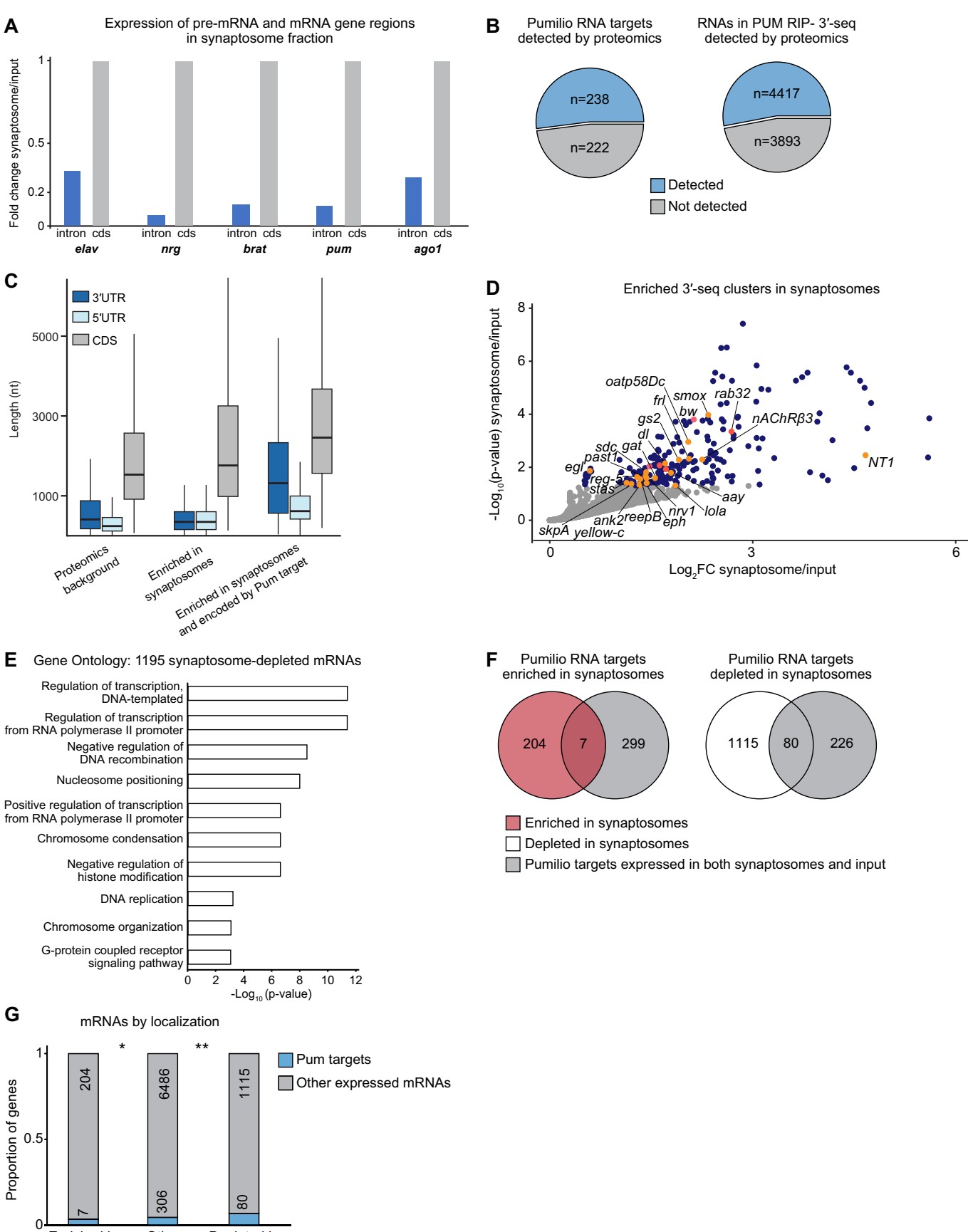

**Figure EV2.  Pum target mRNAs encode proteins enriched in synaptosome fractions.**

(A) RT-qPCR quantification of the indicated transcript regions in synaptosome fractions relative to input fraction. For each gene, intron levels were normalized to coding exon mRNA levels, which were set to the value 1. Ratios represent the average of two biological replicates. (B) Proportions of proteins encoded by Pum target mRNAs (left) and all RNAs identified in Pum xRIP-3′-seq experiment (right) that were detected by proteomics in the synaptosome isolation experiment. (C) Average length of CDS, 5′ UTR and 3′ UTR, of mRNAs encoding proteins of the indicated categories. Number of genes analyzed: $n = 4843$ (Proteomics background), $n = 989$ (enriched in synaptosomes) and $n = 116$ (enriched in synaptosomes and encoded by a Pum target gene). Boxes indicate range between minimum and maximum, the central line depicts the median, lower and upper bounds represent the first and third quartiles, respectively. (D) Differential mRNA expression (by 3′-seq cluster expression) in synaptosome fractions compared to input. The *p*-value of the enrichment is represented as a function of $\log_2$ fold change. Dark blue represents |$\log_2$ fold change (synaptosome/input)| >0 and *p*-value < 0.05 (Wald test). (E) Gene ontology analysis of 1195 mRNAs depleted in synaptosome fractions compared to input. The top ten terms are shown ($p < 0.01$; one-sided EASE score adjusted using the Benjamini-Hochberg method). See Dataset EV3 for all significant terms. (F) Venn diagram showing the intersection between Pum target mRNAs and mRNAs enriched (left) or depleted (right) in the synaptosome fraction. (G) Number and proportion of Pum target mRNAs in each category of mRNA subcellular localization. *$p = 0.1$, **$p < 0.05$ ($p = 4e{-}9$) (two-tailed Fisher's exact test). Only Pum target mRNAs expressed in both synaptosome and input (306 genes) were considered.

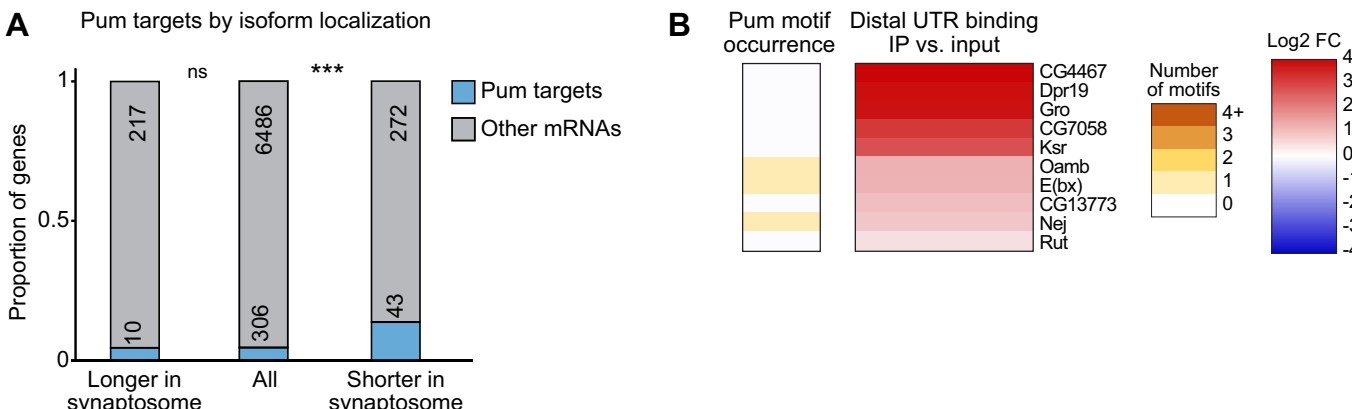

**Figure EV3.  Synaptic localization of short 3′ UTR isoforms of Pumilio target mRNAs.**

(A) Number and proportion of Pum target genes in each category of 3′ UTR isoform subcellular localization. ns, non-significant, ***$p$ < 0.001 ($p$ = 4e−10, two-tailed Fisher's exact test). Only Pum target mRNAs expressed in both synaptosome and input (306 genes) were considered. (B) Heatmaps showing Pum binding and the number of Pum binding motifs in distal 3′ UTR regions of Pum target mRNAs that display longer 3′ UTR isoforms in synaptosomes (10 genes), ranked by Pum xRIP-3′-seq signal compared to input.

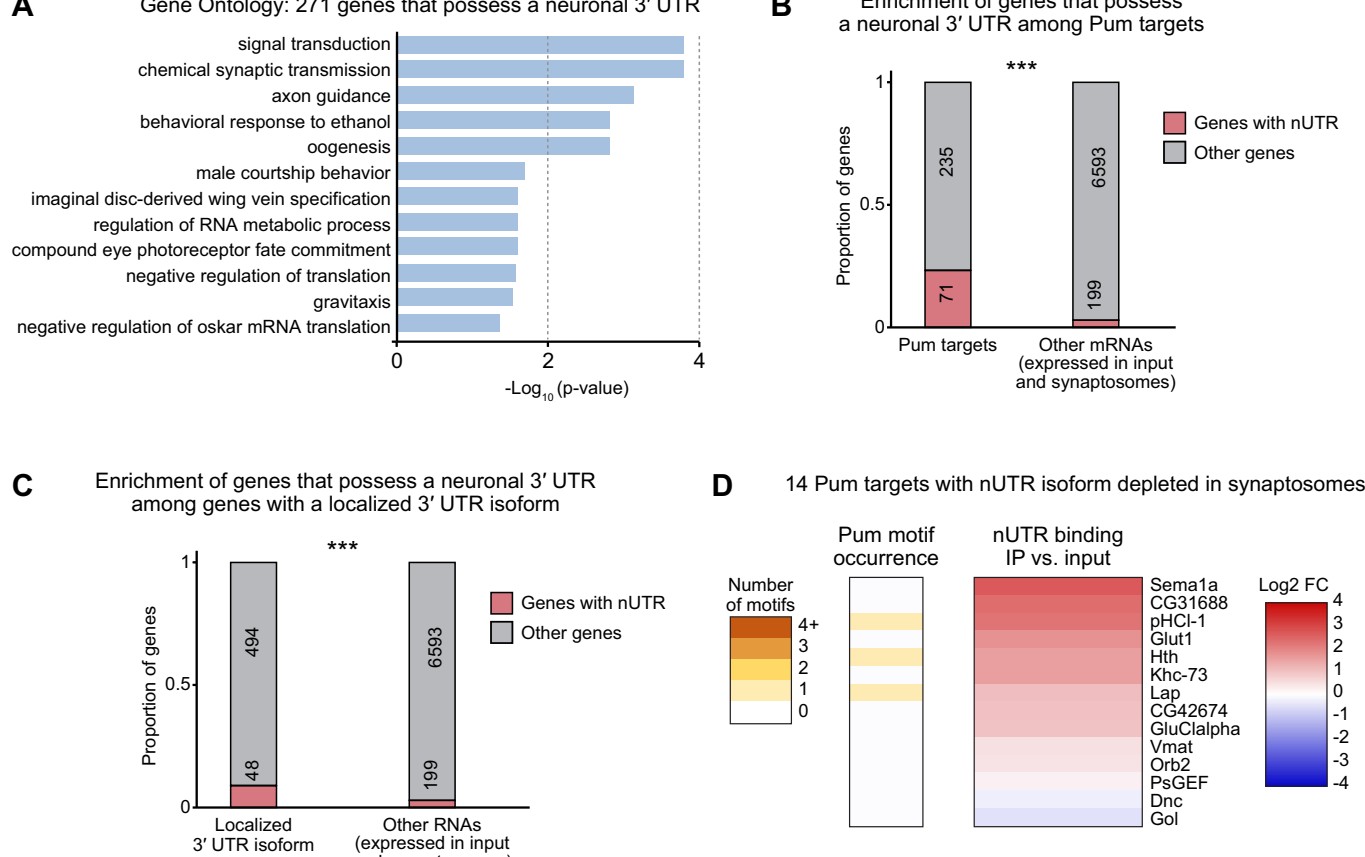

**Figure EV4.  Pumilio binds to soma-localized long neuronal 3′ UTRs of synaptic genes.**

(A) Gene ontology analysis of 271 genes that possess a neuronal 3′ UTR. All significant terms are shown ($p < 0.05$, one-sided EASE score adjusted using the Benjamini-Hochberg method). See also Dataset EV4. (B) Number and proportion of nUTR-containing genes in each gene category. ***$p < 0.001$ ($p = 2e−44$, two-tailed Fisher's exact test). Only mRNAs expressed in both the synaptosome and input samples (199 nUTR-containing genes) were considered. (C) Number and proportion of nUTR-containing genes in each category of 3′ UTR isoform subcellular localization. ***$p < 0.001$ ($p = 2e−10$, two-tailed Fisher's exact test). Only mRNAs expressed in both the synaptosome and input samples (199 nUTR-containing genes) were considered. (D) Heatmaps showing Pum binding and the number of Pum binding motifs in the nUTR of Pum target mRNAs in genes whose nUTR-containing 3′ UTR isoform is depleted in synaptosome fractions (14 genes), ranked by Pum xRIP-3′-seq signal compared to input.

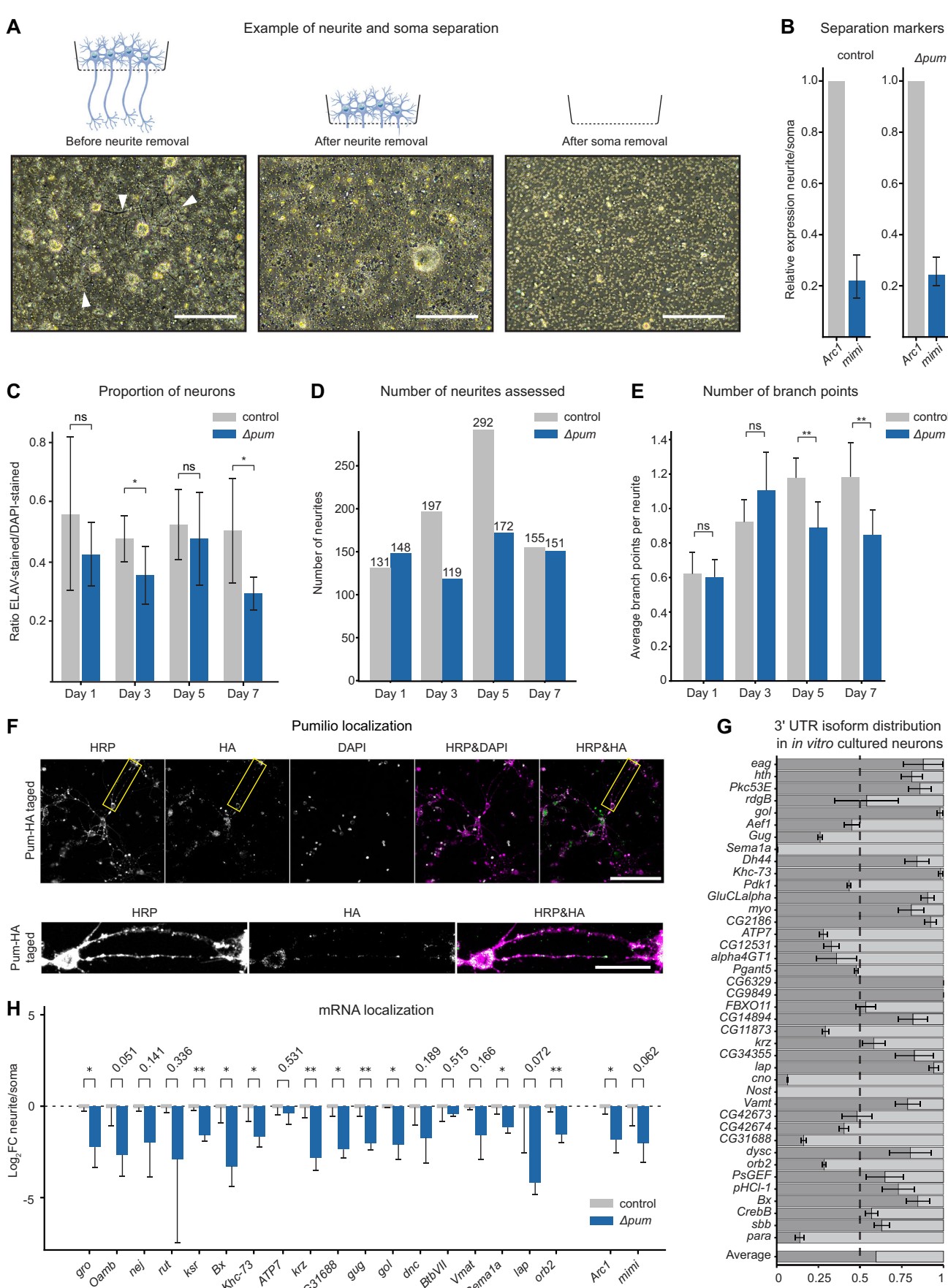

◀ **Figure EV5.** Impaired neurite outgrowth, mRNA delocalization, and synaptic protein overexpression in neurons of *Δpum* flies.

(A) Light micrographs of Drosophila primary neuronal cells cultured on a microporous membrane for separation of neurite and soma compartments. Before removal, neurites are visible as dark protrusions (white arrowheads) below the cell bodies. After soma removal, only cell debris and micropores are visible on the membrane. The three images represent three distinct cultures at 7 days in vitro. Scale bars: 200 μm. (B) Assessment of soma/neurite separation by RT-qPCR quantification of two RNAs well-known to localize to distinct neuronal compartments (neurite/synapse and soma for *Arc1* and *mimi*, respectively). Shown is the RNA expression in neurites relative to cell bodies, in 7 days in vitro cultured neurons of control (genotype: $w^{1118}$) and *Δpum* (genotype: $pum^{ET7}/pum^{ET9}$) flies. For each gene, RNA levels were normalized pairwise, first to soma, and second to *Arc1*. Error bars represent the mean ± SD of three biological replicates for each genotype. (C) Quantification of the proportion of neurons in primary cultures from control ($w^{1118}$) and *Δpum* ($pum^{ET7}/pum^{ET9}$) dissected larval brains at the indicated days in vitro. The number of cells with ELAV staining (a marker of neuronal nuclei) was counted relative to the total number of cells with DAPI staining. Error bars represent the mean ± SD of at least 175 DAPI stained cells for each genotype and time point. *$p < 0.05$, p(3DIV control vs. 3DIV *Δpum*) = 0.03, p(7DIV control vs. 7DIV *Δpum*) = 0.04 (one-tailed Student's t-test). ns, not significant. Total number of cells quantified $n = 2100$. (D) Number of neurites assessed in each genotype and time point for the quantifications shown in Figs. 5C and EV5E. Total number of neurites quantified $n = 1365$. (E) Quantification of average number of branch points per neurite in cultured neurons of control ($w^{1118}$) and *Δpum* ($pum^{ET7}/pum^{ET9}$) flies at the indicated days after plating. Error bars represent the mean ± SD of at least 119 neurites for each genotype and time point. **$p < 0.01$; p(5DIV control vs. 5DIV *Δpum*) = 0.003, p(7DIV control vs. 7DIV *Δpum*) = 0.008 (two-tailed Student's t-test). ns, not significant. Total number of neurites quantified $n = 1365$. (F) Confocal imaging of C-terminally Flag-HA tagged flies ($pum^{Flag}$) flies at 7 days in vitro. HRP marks neuronal membranes. In merged images: HRP (magenta), HA (green) and DAPI (white). Yellow rectangles demarcate the region shown magnified in the lower panels. Scale bars: 50 μm (upper panel), 10 μm (lower panel). (G) Total RNA-seq quantification of long (dark gray) and short (light gray) 3′ UTR isoforms in seven days in vitro cultured neurons of control ($w^{1118}$) flies. 40 genes displaying shorter 3′ UTR isoforms in synaptosomes (longer 3′ UTR isoforms in input) from Fig. 3E are shown (3 genes not detected). Error bars represent the mean ± SD of seven biological replicates. (H) RT-qPCR quantification of the indicated transcripts in neurites relative to cell bodies, in 7 days in vitro cultured neurons of control ($w^{1118}$) and *Δpum* ($pum^{ET7}/pum^{ET9}$) flies. For each gene, RNA levels were normalized to those in control flies (in which $\log_2$ fold change = 0). Error bars represent the mean ± SD of three biological replicates for each genotype. *p*-values are indicated; **$p < 0.01$, *$p < 0.05$; p(*gro* control vs. *gro Δpum*) = 0.02, p(*ksr* control vs. *ksr Δpum*) = 0.004, p(*Bx* control vs. *Bx Δpum*) = 0.012, p(*Khc-73* control vs. *Khc-73 Δpum*) = 0.045, p(*krz* control vs. *krz Δpum*) = 0.002, p(*CG31688* control vs. *CG31688 Δpum*) = 0.02, p(*gug* control vs. *gug Δpum*) = 0.004, p(*gro* control vs. *gro Δpum*) = 2.166e−4, p(*gol* control vs. *gol Δpum*) = 0.047, p(*gro* control vs. *gro Δpum*) = 2.166e−4, p(*Sema1a* control vs. *Sema1a Δpum*) = 0.025, p(*orb2* control vs. *orb2 Δpum*) = 0.007, p(*Arc1* control vs. *Arc1 Δpum*) =0.015 (two-tailed Student's t-test).

