## [Peer Review File · EMBO Reports]

Pumilio differentially binds to mRNA 3' UTR isoforms to regulate localization of synaptic proteins

Dominika Grzejda, Anton Heß, Andrew Rezansoff, Sakshi Gorey, Judit Carrasco, Carlos Alfonso-Gonzalez, Stylianos Tsagkris, Lena Neuhaus, Mengjin Shi, Hasan Can Ozbulut, F.-Nora Voegtle, Andreas Vlachos, and Valerie Hilgers

Corresponding author(s): Valerie Hilgers (hilgers@ie-freiburg.mpg.de)

Review Timeline:

Submission Date:	18th Jul 24
Editorial Decision:	17th Sep 24
Revision Received:	16th Jan 25
Editorial Decision:	30th Jan 25
Revision Received:	5th Feb 25
Accepted:	7th Feb 25

Editor: Esther Schnapp

Transaction Report:

Dear Valerie,

Thank you for your patience while your manuscript was peer-reviewed at EMBO reports. We have finally received the full set of referee reports that is pasted below.

As you will see, the referees acknowledge that the findings are potentially interesting and a good fit for EMBO reports. However, they also have some suggestions for how the study could be improved, and I think all suggestions are good and should be addressed. I had similar thoughts as referee 2 that some of the statements are counterintuitive when I read the ms. Please let me know if you have any comments or questions regarding the revisions and we can discuss the exact requirements further, also in a video chat, if you like.

I would thus like to invite you to revise your manuscript with the understanding that the referee concerns must be fully addressed and their suggestions taken on board. Please address all referee concerns in a complete point-by-point response. Acceptance of the manuscript will depend on a positive outcome of a second round of review. It is EMBO reports policy to allow a single round of major revision only and acceptance or rejection of the manuscript will therefore depend on the completeness of your responses included in the next, final version of the manuscript.

We realize that it is difficult to revise to a specific deadline. In the interest of protecting the conceptual advance provided by the work, we recommend a revision within 3 months (18th Dec 2024). Please discuss the revision progress ahead of this time with the editor if you require more time to complete the revisions.

- 1) A data availability section providing access to data deposited in public databases is missing. If you have not deposited any data, please add a sentence to the data availability section that explains that.
- 2) Your manuscript contains statistics and error bars based on $n=2$. Please use scatter blots in these cases. No statistics should be calculated if $n=2$.

3) We replaced Supplementary Information with Expanded View (EV) Figures and Tables that are collapsible/expandable online. A maximum of 5 EV Figures can be typeset. EV Figures should be cited as 'Figure EV1, Figure EV2' etc... in the text and their respective legends should be included in the main text after the legends of regular figures.

5) a complete author checklist, which you can download from our author guidelines . Please insert information in the checklist that is also reflected in the manuscript. The completed author checklist will also be part of the RPF.

6) Please note that all corresponding authors are required to supply an ORCID ID for their name upon submission of a revised manuscript (. Please find instructions on how to link your ORCID ID to your account in our manuscript tracking system in our Author guidelines

7) Before submitting your revision, primary datasets produced in this study need to be deposited in an appropriate public database (see <https://www.embopress.org/page/journal/14693178/authorguide#datadeposition>). Please remember to provide a reviewer password if the datasets are not yet public. The accession numbers and database should be listed in a formal "Data Availability" section placed after Materials & Method (see also <https://www.embopress.org/page/journal/14693178/authorguide#datadeposition>). Please note that the Data Availability Section is restricted to new primary data that are part of this study. * Note - All links should resolve to a page where the data can be accessed. *
If your study has not produced novel datasets, please mention this fact in the Data Availability Section.

- the name of the statistical test used to generate error bars and P values,
- the number (n) of independent experiments (please specify technical or biological replicates) underlying each data point,
- the nature of the bars and error bars (s.d., s.e.m.),
- If the data are obtained from n {less than or equal to} 2, use scatter blots showing the individual data points.

12) All Materials and Methods need to be described in the main text using our 'Structured Methods' format, which is required for all research articles. According to this format, the Methods section includes a Reagents and Tools Table (listing key reagents, experimental models, software and relevant equipment and including their sources and relevant identifiers) and a Methods and Protocols section describing the methods using a step-by-step protocol format. The aim is to facilitate adoption of the methodologies across labs. More information on how to adhere to this format as well as a downloadable template (.docx) for the Reagents and Tools Table can be found in our author guidelines:
<https://www.embopress.org/page/journal/14693178/authorguide#structuredmethods>.

An example of a Method paper with Structured Methods can be found here: <https://www.embopress.org/doi/full/10.1038/s44320-024-00037-6#sec-4>.

I look forward to seeing a revised form of your manuscript when it is ready.

Best wishes,
Esther

Referee #1:

In this manuscript, Grzejda et al investigate the role of the RNA binding protein Pumilio in regulating the spatial expression of synaptic genes in flies. Their data support that Pumilio preferentially binds to long 3'UTR-transcripts to promote their retention in the soma and repress the expression of the encoded proteins at the synapses.

I found the study interesting and well suited for the journal. Nevertheless, I have some comments to improve the manuscript:

Major comments:

1. The integration of the Pum-RIP, isoform sub-cellular localisation and the 3'UTR length data could be more comprehensive.
 - a. Instead of displaying the comparison 2 by 2, I would suggest having unique representation to displaying the 3 variables using for instance bubble chart, successive heat map etc
 - b. In panel S2C, the authors show that Pum binds more to long 3'UTR. This is an interesting piece of information, but it would be even more informative to compare the binding the level of Pum in the long 3'UTR versus short 3'UTR of transcripts arising from the same gene. Specifically, in figure 2F, showing that the binding level of Pum for the short 3'UTR isoforms is lower than for the long 3'UTR would be highly valuable to support their model.
2. Figure 5: The approach employed to test the causality of Pumilio on the spatial expression of synaptic proteins exhibit quite some confounding effects. The causality of Pumilio binding to long 3'UTR to promote the retention of the transcripts in the soma and subsequent protein expression regulation needs to be further tested experimentally. For instance, the authors could focus on 1-2 candidate(s) and use a reporter strategy with short versus long UTRs, and mutation of Pum binding sites to compare the expression level and localisation of associated transcripts and encoded proteins.
3. The model of the authors would support a somatic localisation of Pumilio protein. Can the authors show it?

Minor comments:

1. Semantic: The expression "Pum target" is confusing. It is not always straightforward to me if the authors refer to the target gene (no matter which transcript isoforms) or to a specific transcript isoform. This would be helpful for the readers to clarify this with 2 different expressions.
2. Table S2C: displaying it as a graph with standard deviations would be easier to read and more informative.
3. Figure 2H: for the top RNA: should not it be "long 3'UTR" (instead of short 3'UTR) and short 3'UTR for the transcript below
4. Figure 5E: The data are quite correlative. To properly interpret the conclusion, localisation and expression levels of the respective transcripts should be analysed.

Referee #2:

In this manuscript, Grzejda and colleagues report that *Drosophila pumilio* protein binds to sites in extended 3' UTRs of several neuronal genes. In doing so, it acts to keep these isoforms out of synaptic compartments. This is important as many of these pumilio targets encode synaptic proteins. These findings complement recently published similar findings in rat neurons where an ortholog of pumilio performs similar functions. Pumilio mutant flies display severe defects in neurite formation as well as aberrant synaptic mislocalization of at least a couple proteins. In general, the conclusions of this manuscript are well-supported by the data. I have a few remaining questions that may help to put these findings in more biological context or make them clearer, but

they are relatively minor.

1. I am struggling a bit to put all the findings together in a way that makes physiological sense for the neuron. For example, in the last line of the middle paragraph of page 8, the authors say "Pum specifically binds the somatically-localized isoforms of this subset of genes [genes with neuron-specific 3' UTR extensions, i.e. nUTR genes]. We propose that by providing a binding platform for Pum, nUTRs promote the localized expression of encoded synaptic proteins."

OK, I understand how Pum is keeping long UTR isoforms in the soma, but how is this promoting localized expression of synaptic proteins? Are the authors suggesting that it is important that these proteins be translated in the soma and then transported to the synapse? I'm OK with that, but if that's the case, it should be clearer. As written, it seems very counterintuitive.

Similarly, in the discussion, the authors say that Pum sites keeping long UTR isoforms in the soma "allows for the proper fate of the short 3' UTR." What does this mean? Why would having a long isoform RNA in the neurite interfere with expression from the short UTR isoform? Is there evidence for this?

2. In these cells, what is the relative abundance of long and short isoforms of the Pum targets? If Pum is acting on only the long isoforms to keep them in the soma, what fraction of the total RNA for that gene is it operating on? How big of an effect will these be for overall expression of the encoded protein (from either isoform)?

3. I would like some more information about how the RT-qPCR experiments (particularly S5F) were done, especially in regards to normalization. Is the same amount of RNA going into every reaction? Or the same volume of lysate? Or is there an external "standard" gene that everything is being compared to?

I understand how impaired neurite formation would lead to less RNA being recovered from the neurite fraction of Pum mutant neurons, but there is likely still information to be gained in such a scenario. Presumably Pum-target RNAs will be more strongly affected than others, so if the quantitative measurement is a relative one (like is done in RNAseq experiments), they might stick out, lending more evidence to Pum directly impacting their localization. For example, if Pum target X is 1% of total neurite RNA in wildtype flies but 2% of total neurite RNA in Pum mutant flies, you might be able to say that this RNA is mislocalized, even if its absolute amount decreased in Pum neurites. It just decreased less than other, non-target RNAs.

Referee #3:

This study investigates RNA-dependent regulation of mRNA localization and translation specifically in neurons. The study uses exclusively *Drosophila* as a model system, but the authors provide reasonable arguments for a broad conservation of the molecular mechanism/s.

The experiments are focused on investigating the function of pumilio, an important RNA binding protein. Using state of the art techniques, the authors present a set of very interesting and in part surprising results, suggesting for example a competitive RNA localization model for mRNA with long- versus short UTRs.

The experiments and results are solid and certainly of broad interest for molecular neuroscientist as well as anybody interested in translational and cellular regulation.

I recommend publication essentially as it is, with only a few comments listed below.

- 1). It might be useful if the authors comment in the text that the mechanistic model they investigate in the first part of the paper (pum selectively binds long 3'UTRs, retaining it in the soma compartment, while corresponding mRNAs isoforms with short 3'UTR encode proteins that get enriched at synapses) is derived from an analysis of adult flies. In contrast, the functional consequences of losing pum are studied in a second part in a developmental paradigm. It seems - at this point - that the pum-dependent regulation in mature versus developing neurons may or may not be the same. I think this should be stated clearly.
- 2) As the authors have generated a FLAG-HA-tagged pum knock-in, I was wondering whether they have used this to actually localize Pum protein distribution in neurons. Is it consistent with a proposed soma concentration of Pum?
- 3). A summary figure explaining the key results and/or the proposed model would be helpful for readers. In the current version it is a bit tedious to establish the full context.

MPI of Immunobiology and Epigenetics • Stuebeweg 51 • D-79108 Freiburg

EMBO Reports

Editorial team

Meyerhofstrasse 1
D-69117 Heidelberg
Germany

Dr. Valérie Hilgers
Max Planck Research Group Leader
Stübeweg 51
79108 Freiburg im Breisgau
Office: +49 761 5108-280
Mobile phone: +49 1522 398-1782
hilgers@ie-freiburg.mpg.de

Freiburg, January 16, 2025

Dear Dr. Schnapp, dear *EMBO Reports* editorial team,

Please find below, a detailed point-by-point response to all of the referees' comments and suggestions for our manuscript titled "Pumilio differentially binds to mRNA 3' UTR isoforms to regulate localization of synaptic proteins" (EMBOR-2024-60013V1).

We are grateful for the reviewers' detailed and helpful suggestions, which we believe have greatly improved our manuscript. Our responses are in blue.

Referee #1:

In this manuscript, Grzejda et al investigate the role of the RNA binding protein Pumilio in regulating the spatial expression of synaptic genes in flies. Their data support that Pumilio preferentially binds to long 3'UTR-transcripts to promote their retention in the soma and repress the expression of the encoded proteins at the synapses.

I found the study interesting and well suited for the journal. Nevertheless, I have some comments to improve the manuscript:

We thank Referee #1 for their positive review and very useful comments.

Major comments:

1. The integration of the Pum-RIP, isoform sub-cellular localisation and the 3'UTR length data could be more comprehensive.

a. Instead of displaying the comparison 2 by 2, I would suggest having unique representation to displaying the 3 variables using for instance bubble chart, successive heat map etc

b. In panel S2C, the authors show that Pum binds more to long 3'UTR. This is an interesting piece of information, but it would be even more informative to compare the binding the level of Pum in the long 3'UTR versus short 3'UTR of transcripts arising from the same gene. Specifically, in figure 2F, showing that the binding level of Pum for the short 3'UTR isoforms is lower than for the long 3'UTR would be highly valuable to support their model.

We thank Referee #1 for their helpful suggestions to better integrate the data.

a. We tried three-way representations to integrate all three datasets but did not succeed in displaying them in a good way. Instead, we removed the confusing plot, and replaced it with a horizontal barplot

(in Fig. 3E of the revised manuscript) representing localization of by 3'UTR length, of RNAs bound by Pum.

b. We performed the requested analysis. We calculated the relative binding of Pumilio in long compared to short 3'UTR isoforms of transcripts arising from the same gene. We find a significant enrichment of the long (compared to short) isoform in the Pum xRIP, but not the control xRIP. The new analysis is described in the methods section and the new data is shown in Fig 3F of the revised manuscript.

2. Figure 5: The approach employed to test the causality of Pumilio on the spatial expression of synaptic proteins exhibit quite some confounding effects. The causality of Pumilio binding to long 3'UTR to promote the retention of the transcripts in the soma and subsequent protein expression regulation needs to be further tested experimentally. For instance, the authors could focus on 1-2 candidate(s) and use a reporter strategy with short versus long UTRs, and mutation of Pum binding sites to compare the expression level and localisation of associated transcripts and encoded proteins.

We agree that the severe neuronal phenotypes of *pum* mutants create confounding effects that may affect spatial expression of synaptic proteins. The same confounding effects prevented us from measuring RNA expression in *pum* mutant neurites. We have tried the reporter strategy suggested by Referee #1. For two genes, we generated transgenic flies with constructs containing either 1) the entire, wild-type 3'UTR (short UTR and neuronal UTR), 2) only the short UTR, 3) the entire 3'UTR with Pum binding sites mutated in the neuronal UTR. Unfortunately, while GFP and shortUTR regions could be detected by FISH and immunostaining, nUTR regions were not expressed from these constructs (constructs 1 and 3), even in neurons under the control of *elav-Gal4*. Ectopically expressing nUTRs has proved very difficult, in our lab and in others. The influence of promoters and enhancers on 3'end site selection could be a possible explanation; although we used the promoter regions of the native genes, the region chosen (+300 bp) may have been insufficient to confer alternative 3'end site selection and nUTR expression.

In the revised version of the manuscript, we clearly state the confounding effects and tone down the claims of causality of Pumilio in the spatial expression of synaptic proteins.

3. The model of the authors would support a somatic localisation of Pumilio protein. Can the authors show it?

We agree that our model supports a somatic localisation of Pumilio protein. As also requested by Referee #3, we tested Pum expression and localisation experimentally. We used an anti-HA antibody in neurons cultured from flies expressing an endogenously tagged Pum protein. We co-stained Pum with DAPI and with a marker for neuronal membranes. Pum was expressed in neurons in culture, and was detectable both in neurites and in soma, with much stronger signal in somata. While we are careful not to over-interpret this as "somatic localisation", our imaging shows that Pum is mainly expressed in the soma of neurons. These data are shown in Fig. EV5F in the revised manuscript.

Minor comments:

1. Semantic: The expression "Pum target" is confusing. It is not always straightforward to me if the authors refer to the target gene (no matter which transcript isoforms) or to a specific transcript isoform. This would be helpful for the readers to clarify this with 2 different expressions.

We apologize for the confusion. To improve clarity, in the revised manuscript text we consistently use the term "Pum target (gene)" when referring to the gene and "Pum target isoform/mRNA" when referring to specific transcripts. We also clarify the use of each term at the beginning of the results section.

2. Table S2C: displaying it as a graph with standard deviations would be easier to read and more informative.

We performed the suggested change. The numbers are now integrated into a box plot with standard deviations. We agree that the data (Fig. EV2C in the revised manuscript) are much better displayed now.

3. Figure 2H: for the top RNA: should not it be "long 3'UTR" (instead of short 3'UTR) and short 3'UTR for the transcript below.

We agree that the labeling of the transcripts was confusing. We changed it according to Referee #1's suggestions. The model is now shown in Fig. 6 of the revised manuscript.

4. Figure 5E: The data are quite correlative. To properly interpret the conclusion, localisation and expression levels of the respective transcripts should be analysed.

We agree that these data are correlative. As mentioned in response to Referee #1's Major comment 2, work on pum mutant cultures was very difficult due to the severity of the neuronal phenotypes in the mutant neurons. We attempted both FISH and qPCR on short and long 3' UTRs of Pum target mRNAs; both approaches yielded uninterpretable effects due to the lack of proper neurites in pum neurons.

In the revised manuscript, we discuss the results with this in mind, and avoid direct claims of causality.

Referee #2:

In this manuscript, Grzejda and colleagues report that *Drosophila pumilio* protein binds to sites in extended 3' UTRs of several neuronal genes. In doing so, it acts to keep these isoforms out of synaptic compartments. This is important as many of these pumilio targets encode synaptic proteins. These findings complement recently published similar findings in rat neurons where an ortholog of pumilio performs similar functions. Pumilio mutant flies display severe defects in neurite formation as well as aberrant synaptic mislocalization of at least a couple proteins. In general, the conclusions of this manuscript are well-supported by the data. I have a few remaining questions that may help to put these findings in more biological context or make them clearer, but they are relatively minor.

We thank Referee #2 for their thoughtful comments, and for helping us improve the clarity of this paper.

1. I am struggling a bit to put all the findings together in a way that makes physiological sense for the neuron. For example, in the last line of the middle paragraph of page 8, the authors say "Pum specifically binds the somatically-localized isoforms of this subset of genes [genes with neuron-specific 3' UTR extensions, i.e. nUTR genes]. We propose that by providing a binding platform for Pum, nUTRs promote the localized expression of encoded synaptic proteins."

OK, I understand how Pum is keeping long UTR isoforms in the soma, but how is this promoting localized expression of synaptic proteins? Are the authors suggesting that it is important that these proteins be translated in the soma and then transported to the synapse? I'm OK with that, but if that's the case, it should be clearer. As written, it seems very counterintuitive.

We apologize for not stating our interpretation and model more clearly. We think that Pum regulates the expression of the synaptic protein by repressing translation of the long isoform in a localized manner, thereby creating a stronger "expression gradient" between soma and neurite.

In the revised manuscript, we rewrite this part, which is now: "We propose that by providing a binding platform for Pum, nUTRs, in the neuronal soma, restrict translation of mRNAs encoding synaptic proteins. Since the short isoform of the same gene can localize to the synapse, nUTRs indirectly promote a more localized expression of synaptic proteins."

While our data indicate, and we propose, that the short isoform is translated at the synapse, we do not provide direct functional evidence of this. We discuss these questions and considerations in the results and discussion sections of the revised version of the manuscript.

Similarly, in the discussion, the authors say that Pum sites keeping long UTR isoforms in the soma "allows for the proper fate of the short 3' UTR." What does this mean? Why would having a long isoform RNA in the neurite interfere with expression from the short UTR isoform? Is there evidence for this?

We apologize for this confusing statement. We amended the discussion text to discuss the effect of Pumilio in a clearer manner.

2. In these cells, what is the relative abundance of long and short isoforms of the Pum targets? If Pum is acting on only the long isoforms to keep them in the soma, what fraction of the total RNA for that gene is it operating on? How big of an effect will these be for overall expression of the encoded protein (from either isoform)?

We performed new experiments to assess the relative abundance of long and short isoforms of Pum target mRNAs. We performed RNA-seq on primary neurons cultured from wild-type *Drosophila* larvae (soma and neurites pooled), and analyzed relative expression of short-UTR vs. nUTR regions. Relative abundances vary substantially from gene to gene; however for most genes, the long isoform represents roughly half (over 50% on average) of the gene's mRNA expression, indicating that nUTRs have a large regulatory potential. The data is displayed in Fig. EV5G, representing the 43 Pum target genes (40 shown, 3 undetected) whose nUTR-containing isoform is enriched in synaptosomes.

3. I would like some more information about how the RT-qPCR experiments (particularly S5F) were done, especially in regards to normalization. Is the same amount of RNA going into every reaction? Or the same volume of lysate? Or is there an external "standard" gene that everything is being compared to?

The same amount of approximately 200 ng total RNA went into each reaction. Soma and neurite fraction that originated from the same replicate were analyzed pairwise for each gene. We also used a mouse spike-in (5%); however, since in the shown plots, neurite/soma ratios (Fig. EV5H) or RNA long/short ratios (Fig. 5D) were calculated, the normalization by spike-in was cancelled out. We added this description in the Methods and Protocols section.

I understand how impaired neurite formation would lead to less RNA being recovered from the neurite fraction of Pum mutant neurons, but there is likely still information to be gained in such a scenario. Presumably Pum-target RNAs will be more strongly affected than others, so if the quantitative measurement is a relative one (like is done in RNAseq experiments), they might stick out, lending more evidence to Pum directly impacting their localization. For example, if Pum target X is 1% of total neurite RNA in wildtype flies but 2% of total neurite RNA in Pum mutant flies, you might be able to say that this RNA is mislocalized, even if its absolute amount decreased in Pum neurites. It just decreased less than other, non-target RNAs.

We thank Referee #2 for this suggestion. We agree that this would be the best way to assess change of localization in the mutant with severely impaired neurites. We did not measure total RNA amounts with quantitative precision, or perform RNA-seq on the samples (which, in hindsight, we should have); the recovered amount of RNA in Δpum neurites was barely sufficient for the few qPCR reactions we show in the figures. We analyzed the data in different ways, including normalizing target RNA qPCR signal to that of a "control" gene for neurites and soma separately, which we consider the solution closest to normalizing to total RNA; however in *pum* mutants, the bias towards soma (shown in Fig. EV5H) for all transcripts (including that of the control mRNAs *Arc1*, a neurite marker, and *mimi*, a soma marker) was seen no matter what type of normalization we tried, which made interpretation of these results difficult.

Referee #3:

This study investigates RNA-dependent regulation of mRNA localization and translation specifically in neurons. The study uses exclusively *Drosophila* as a model system, but the authors provide reasonable arguments for a broad conservation of the molecular mechanism/s.

The experiments are focused on investigating the function of pumilio, an important RNA binding protein. Using state of the art techniques, the authors present a set of very interesting and in part surprising results, suggesting for example a competitive RNA localization model for mRNA with long-versus short UTRs.

The experiments and results are solid and certainly of broad interest for molecular neuroscientists as well as anybody interested in translational and cellular regulation.

I recommend publication essentially as it is, with only a few comments listed below.

We thank Referee #3 for their positive assessment of our paper, and for their very helpful insight.

1). It might be useful if the authors comment in the text that the mechanistic model they investigate in the first part of the paper (pum selectively binds long 3'UTRs, retaining it in the soma compartment, while corresponding mRNAs isoforms with short 3'UTR encode proteins that get enriched at synapses) is derived from an analysis of adult flies. In contrast, the functional consequences of losing pum are

studied in a second part in a developmental paradigm. It seems - at this point - that the pum-dependent regulation in mature versus developing neurons may or may not be the same. I think this should be stated clearly.

We thank Referee #3 for bringing up this important consideration. In the revised manuscript, we added a sentence to clearly state the possibility that the effects we observe in Δ pum cultured neurons may be specific to developing flies: "Although these observations, stemming from developing neurons cultured *in vitro*, may not fully reflect RNA processes that occur in the adult brain, they are consistent with our mechanistic model."

2). As the authors have generated a FLAG-HA-tagged pum knock-in, I was wondering whether they have used this to actually localize Pum protein distribution in neurons. Is it consistent with a proposed soma concentration of Pum?

We thank Referee #3 for raising this excellent point. As also requested by Referee #1, we performed immunofluorescence imaging on neurons cultured from *pum^{Flag}* flies (the FLAG-HA Pum knock-in) using an anti-HA antibody to assess Pum protein distribution in neurons. We imaged neurons at 7 days *in vitro* (the system shown in Figures 5 and EV5). We found tagged Pumilio both in neurites and in soma, with much stronger signal in cell bodies, which is consistent with the proposed soma concentration of Pum *in vivo*. These data are shown in Fig. EV5F in the revised manuscript.

3). A summary figure explaining the key results and/or the proposed model would be helpful for readers. In the current version it is a bit tedious to establish the full context.

In the revised manuscript, we include a better summary figure. We improved the clarity of the model (previously Fig. 2H) and moved it as a stand-alone panel into the last figure (Fig. 6), where we think it is most helpful as a visual summary for the readers. In addition, we prepared a graphical abstract with the key results and model.

We wish to thank once again, the three Reviewers for very constructive reviews and are immensely grateful for the opportunity to address their concerns in this revised manuscript. Please do not hesitate to contact me if you have any questions or concerns.

Sincerely,

Dr. Valérie Hilgers

Max Planck Research Group Leader

Dear Valerie,

Thank you for the submission of your revised manuscript. We have now received the enclosed reports from the referees and I am happy to say that all support its publication now. Only a few editorial requests will need to be addressed before we can proceed with the official acceptance of your manuscript.

- Please remove the author credits from the ms file. All credits need to be entered during online ms submission.
- Please add this FUNDING INFO - Max Planck Society - when you upload the final ms into our online submission system.
- A Fig. S5E is called out in the ms text and needs to be updated to the correct figure callout.
- The 5 EV tables need to be called Datasets since they have multiple tabs; the correct nomenclature (in the source file names, titles in legends and ms callouts) should be Dataset EV1-EV5; and their legends need to be removed from the ms file (it is sufficient to provide them in each Excel file).
- Please remove the instructions and example from the Reagents and Tools table file.
- Methods and Protocols needs to be corrected to "Methods"

Figure Legends - Comments

- Please note that the exact p values are not provided in the legends of figures 2E, G; 3D, F; 4B, D; 5C, D; EV2 G, EV3 A; EV4 B, C; EV5 C, E, H. Please provide exact p-values as reasonable.
- Please indicate the statistical test used for data analysis in the legends of figures 1D, F; 2D, F; 4C, D; EV2 D, E; EV4 A.
- Please indicate what */ **/ ***/ **** represents; if this represents p value(s), please specify exact p value in the legend(s) of figure(s) 4C.
- Please note that the box plots need to be defined in terms of minima, maxima, centre, bounds of box and whiskers, and percentile in the legends of figures 3F, 5E, EV2 C
- Please note that information related to n is missing in the legend of figure 3F.
- Please note that the red border is not defined in the legend of figure 2B. This needs to be rectified.

I would like to suggest some minor changes to the abstract that needs to be written in present tense:

In neuronal cells, the regulation of RNA is crucial for the spatiotemporal control of gene expression, but how the correct localization, levels, and function of synaptic proteins are achieved is not well understood. In this study, we globally investigate the role of alternative 3' UTRs in regulating RNA localization in the synaptic regions of the Drosophila brain. We identify direct mRNA targets of the translational repressor Pumilio, finding that mRNAs bound by Pumilio encode proteins enriched in synaptosomes. Pumilio differentially binds to RNA isoforms of the same gene, favoring long, neuronal 3' UTRs. These longer 3' UTRs tend to remain in the neuronal soma, whereas shorter UTR isoforms localize to the synapse. In cultured pumilio mutant neurons, axon outgrowth defects are accompanied by mRNA isoform mislocalization, and proteins encoded by these Pumilio target mRNAs display excessive abundance at synaptic boutons. Our study identifies an important mechanism for the spatiotemporal regulation of protein function in neurons.

[If you would like to keep "widespread" please explain what you mean by that.]

I also slightly modified the last bullet point. Do you agree with:

- Pumilio preferentially binds long 3' UTR isoforms to restrict expression of synaptic protein isoforms to neuronal in cell bodies

The quality of the synopsis image could be improved. I attach the image at its final size to this email, the text could be larger. If you send us an image file (tif or jpg) at its final size of 550 pixels x 380 pixels and may be increase the font it will probably look better.

Best wishes,
Esther

Referee #1:

The authors have addressed all my points.
I fully support the publication of this nice manuscript at EMBO Report.

Referee #2:

The authors have satisfactorily addressed my concerns.

All editorial and formatting issues were resolved by the authors.

Valerie Hilgers
Max Planck Institute of Immunobiology and Epigenetics
Stuebeweg 51
Freiburg, Baden-Württemberg 79108
Germany

Dear Valerie,

I am very pleased to accept your manuscript for publication in the next available issue of EMBO reports. Thank you for your contribution to our journal.

Best,
Esther
